# Neutrophil stunning by metoprolol reduces infarct size

Jaime García-Prieto[1,2], Rocío Villena-Gutiérrez[1], Mónica Gómez[1], Esther Bernardo[3], Andrés Pun-García[1], Inés García-Lunar[1,2,4,5], Georgiana Crainiciuc[1], Rodrigo Fernández-Jiménez[1,2,3], Vinatha Sreeramkumar[1,4], Rafael Bourio-Martínez[1,6], José M. García-Ruiz[1,2,7], Alfonso Serrano del Valle[1], David Sanz-Rosa[1,2,4], Gonzalo Pizarro[1,2,4,8], Antonio Fernández-Ortiz[1,2,3], Andrés Hidalgo[1,9], Valentín Fuster[1,10] & Borja Ibanez[1,2,11]

The β1-adrenergic-receptor (ADRB1) antagonist metoprolol reduces infarct size in acute myocardial infarction (AMI) patients. The prevailing view has been that metoprolol acts mainly on cardiomyocytes. Here, we demonstrate that metoprolol reduces reperfusion injury by targeting the haematopoietic compartment. Metoprolol inhibits neutrophil migration in an ADRB1-dependent manner. Metoprolol acts during early phases of neutrophil recruitment by impairing structural and functional rearrangements needed for productive engagement of circulating platelets, resulting in erratic intravascular dynamics and blunted inflammation. Depletion of neutrophils, ablation of *Adrb1* in haematopoietic cells, or blockade of PSGL-1, the receptor involved in neutrophil–platelet interactions, fully abrogated metoprolol's infarct-limiting effects. The association between neutrophil count and microvascular obstruction is abolished in metoprolol-treated AMI patients. Metoprolol inhibits neutrophil–platelet interactions in AMI patients by targeting neutrophils. Identification of the relevant role of ADRB1 in haematopoietic cells during acute injury and the protective role upon its modulation offers potential for developing new therapeutic strategies.

[1] Centro Nacional de Investigaciones Cardiovasculares Carlos III (CNIC), 28029 Madrid, Spain. [2] CIBER de enfermedades CardioVasculares (CIBERCV), 28029 Madrid, Spain. [3] Hospital Clínico San Carlos, 28040 Madrid, Spain. [4] Clinical Department, School of Biomedical Sciences, Universidad Europea, 28670 Madrid, Spain. [5] Hospital Universitario Quirón, 28223 Madrid, Spain. [6] Hospital de Basurto, 48013 Bilbao, Spain. [7] Hospital Universitario Central de Asturias (HUCA), 33011 Oviedo, Spain. [8] Complejo Hospitalario Ruber Juan Bravo-UEM, 28006 Madrid, Spain. [9] Institute for Cardiovascular Prevention (IPEK), Ludwig-Maximilians University, 80336 Munich, Germany. [10] Zena and Michael A. Wiener Cardiovascular Institute, Icahn School of medicine at Mount Sinai, New York, New York 10029, USA. [11] Department of Cardiology, Instituto de Investigación Sanitaria (IIS)-Fundación Jiménez Díaz, 28040 Madrid, Spain. Correspondence and requests for materials should be addressed to B.I. (email: bibanez@cnic.es).

Heart attack (acute myocardial infarction, AMI) is one of the principal manifestations of cardiovascular disease and a chief contributor to mortality and morbidity worldwide. The main determinant of poor outcome post AMI is the extent of irreversible injury (infarct size). The mainstay of AMI treatment is rapid reperfusion to restore blood flow, which reduces complications and improves survival. However, reperfusion itself accelerates and exacerbates the inflammatory response associated with myocardial injury[1]. Thus, the injury inflicted on the myocardium during AMI is the result of ischaemia and reperfusion processes, and is known as ischaemia/reperfusion (IR) injury. The development of effective therapies to reduce myocardial IR injury is an unmet clinical need[2].

The injured myocardium is infiltrated by circulating neutrophils, and these cells are critically involved in myocardial IR injury[3–7]. In an inflammatory milieu, neutrophils bind to platelets and red blood cells, forming plugs[7]. Upon reperfusion, these plugs are dispersed into the microcirculation, where they form embolisms, precluding tissue perfusion despite blood flow restoration in the large coronary arteries. This phenomenon, known as microvascular obstruction (MVO), is a major contributor to IR injury and infarct size[1]. Moreover, neutrophil infiltration into acutely damaged organs is dependent on their interaction with platelets[8], and these interactions are critical to the formation of harmful co-aggregates and the initiation of inflammatory-like responses before tissue infiltration[3,8]. Overall, neutrophil dynamics (including neutrophil–platelet interactions) are an attractive therapeutic target for the prevention of IR injury.

The intravenous (i.v.) administration of the selective β1-adrenergic receptor (ADRB1) antagonist metoprolol has been shown to reduce infarct size and improve long-term cardiac function after AMI in the recent METOCARD-CNIC trial[9,10]. The mechanism underlying metoprolol's cardioprotective effect remains unclear[11]. Identifying this mechanism could have significant implications for the understanding of IR and the development of novel infarct-limiting therapies. The adrenergic system is critically involved in inflammatory reactions[12,13]. In particular, the inflammatory response of neutrophils involves the *de novo* production and release by these cells of catecholamines[12,13]. Induced catecholamine stress (as during ischaemia) alters neutrophil trafficking[14–16] and promotes formation of neutrophil–platelet co-aggregates[17].

We hypothesized that pre-reperfusion i.v. metoprolol administration alters neutrophil dynamics, resulting in a dampened inflammatory response, less severe reperfusion injury and smaller infarcts. Here we show that pre-reperfusion administration of i.v. metoprolol to AMI patients significantly reduces the incidence of MVO, and moreover that metoprolol inhibits deleterious neutrophil inflammatory responses both in patients and in animal models of IR. The infarct-limiting effect of metoprolol is abolished in neutrophil-depleted mice and when neutrophils are prevented from interacting with platelets. The beneficial effects of metoprolol are also abolished by genetic ablation of *Adrb1* and are rescued by restitution of *Adrb1* expression only in haematopoietic cells. These results identify the neutrophil dynamics as the target of the cardioprotective effect of metoprolol against myocardial IR injury.

## Results

**Intravenous metoprolol reduces MVO in AMI patients**. The METOCARD-CNIC trial recruited patients with an ongoing acute ST-segment elevation AMI and randomized them to receive i.v. metoprolol (15 mg) or control before reperfusion[18]. A total of 220 AMI patients underwent a cardiac magnetic resonance (CMR) imaging exam 1 week after AMI. To study the potential mechanisms underlying the infarct-limiting effect of metoprolol[9], we analysed the 1-week CMR data to evaluate the extent of MVO. MVO was defined as the absence of contrast wash-in inside the delayed gadolinium-enhanced area[19], and was quantified as grams of left ventricle (LV) (Fig. 1a,b and Supplementary Fig 1a,b). Patients treated with metoprolol during ongoing AMI had a 40% lower extent of MVO (Fig. 1c). This significant effect was maintained after adjusting for factors potentially affecting MVO by performing linear multiple regression analysis and including sex, age, ischaemia duration, diabetes, and use of thrombectomy or glycoprotein IIb/IIIa inhibitors as covariates. To exclude the possibility that this effect simply reflected the reduction in total infarct size[9], MVO was further assessed as a percentage of the infarcted area (total late gadolinium enhanced area). Metoprolol-treated patients had 24% less infarct-normalized MVO than control patients (Fig. 1d). As expected, the extent of MVO was significantly associated with poor long-term outcome, evaluated as chronic ventricular performance (Supplementary Fig. 1c). These data suggest that MVO reduction might be involved in the cardioprotective effect of metoprolol administration in patients during ongoing AMI.

**Intravenous metoprolol dampens neutrophilial–MVO association in AMI**. White blood cell (WBC) and neutrophil counts during an AMI are known to be associated with larger infarct sizes and extensive MVO[20–22]. We explored these associations in AMI patients from the METOCARD-CNIC trial. We found a significant positive correlation between absolute leukocyte count on admission and the extent of MVO on CMR: the higher the leukocyte count, the larger the extent of MVO (Fig. 2a). We further studied the association of the different WBC subpopulations and MVO. As expected, neutrophil count was significantly correlated with the extent of MVO (Fig. 2b). Conversely, there was no sign of association between other WBC subpopulations and MVO: lymphocyte, monocyte, eosinophil or platelet counts did not correlate with the extent of MVO (Fig. 2e). Next, we studied the effect of metoprolol on WBCs and on the association between these and MVO. Metoprolol treatment was not associated with any different in WBC count nor in any WBC subpopulation (Supplementary Table 1). Of note, we found a significant interaction between metoprolol treatment and the correlation between leukocyte count and MVO: the significant positive correlation between neutrophil count and the extent of MVO was only present in control patients (that is, not receiving metoprolol); in patients receiving i.v. metoprolol before reperfusion there was no sign of association between total leukocyte or neutrophil counts and the extent of MVO (Fig. 2c,d). These results suggest that the administration of i.v. metoprolol during ongoing AMI do not affect the circulating levels of WBCs but modulates the impact of neutrophils on MVO.

**Metoprolol blunts neutrophil infiltration in experimental IR**. To identify the factors underlying the metoprolol-induced reduction in MVO within the reperfused myocardium, we examined a mouse model of *in vivo* myocardial IR injury (Fig. 3a). Given the observed modulator effect of i.v. metoprolol in the association between neutrophil count and MVO in patients suffering an AMI, we focused our attention into this cell population. We first tested the infarct-limiting effect of metoprolol in wild-type mice by occluding the left anterior descending (LAD) coronary artery for 45 min followed by reperfusion. Mice were randomized to receive a single i.v. bolus (50 μl) of metoprolol (10 mM) or vehicle (saline) 35 min after

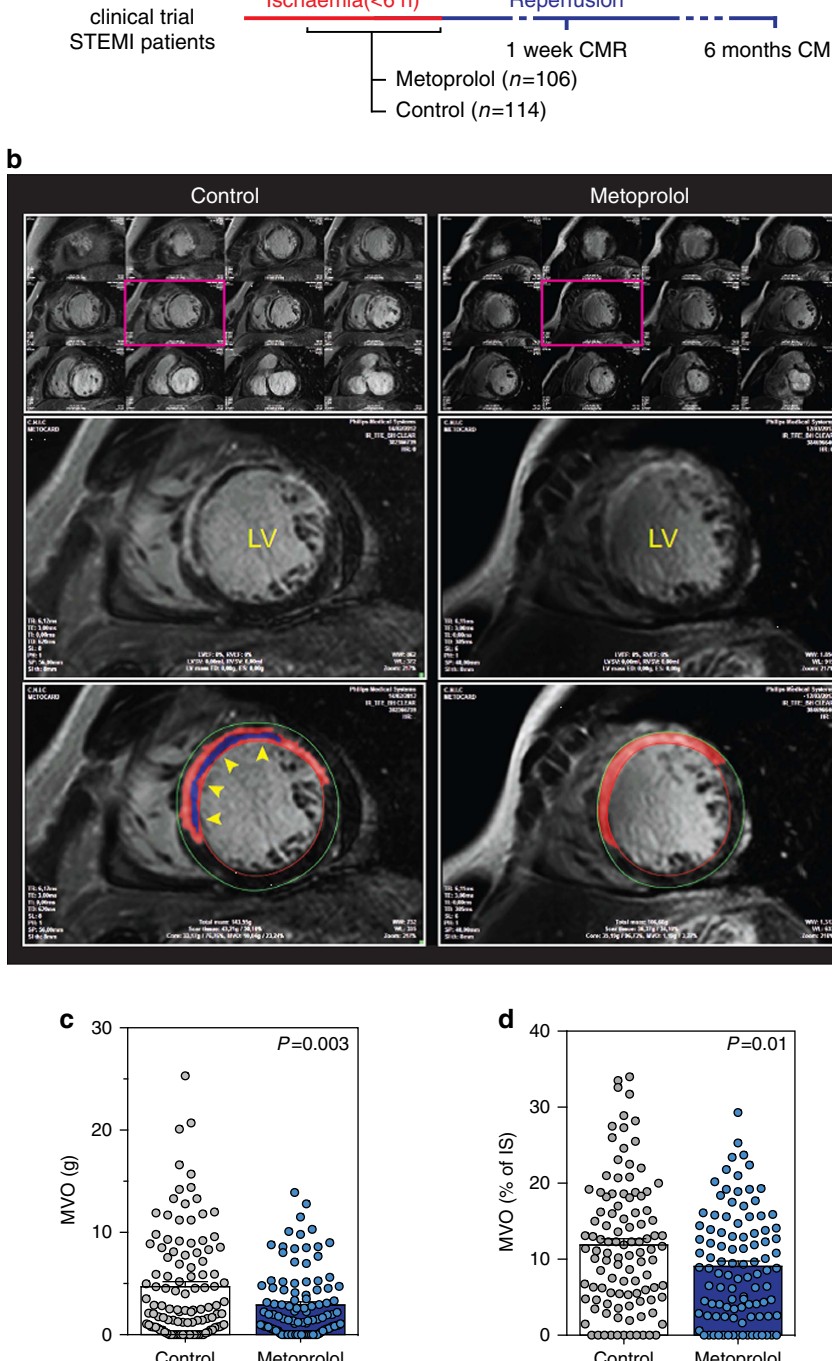

**Figure 1 | Metoprolol administration during ongoing AMI reduces MVO in patients.** (**a**) METOCARD-CNIC trial scheme: patients with ongoing ST-segment elevation myocardial infarction (STEMI) were recruited and randomized to receive metoprolol (15 mg i.v. doses) or control before reperfusion. A total of 220 patients were evaluated for MVO by cardiac magnetic resonance (CMR) imaging 1 week after AMI and 202 patients for an additional CMR at long-term LVEF 6 months after AMI for ventricular function assessment. (**b**) Representative CMR exams (short-axis covering the entire left ventricle (LV) from base to apex), showing significant differences in 1-week MVO between a control patient (left) and a metoprolol-treated patient (right). Lower panels show detailed views of the boxed images, revealing MVO (blue area, automatic quantification), defined as the absence of contrast wash-in inside the delayed gadolinium-enhanced area (red, semiautomatic quantification). Yellow arrowheads indicate MVO in the LV wall. (**c**) Quantification of MVO in grams of left ventricle. (**d**) Quantification of MVO relative to the infarcted area (%). Dots represent values for individual patients: 114 in the control group (grey) versus 106 in the metoprolol group (blue). Data are means ± s.e.m. and compared by unpaired Student's *t*-test.

ischaemia onset (10 min before reperfusion). Infarct size was evaluated at 24 h reperfusion by TTC staining and normalized to area at risk (AAR, negative Evans blue staining). Metoprolol-treated mice had significantly smaller infarcts (as % of AAR) than vehicle-treated mice (Fig. 3b–d). Next, we examined mice carrying a GFP reporter in myeloid derived cells (LysM-GFP). LysM-GFP mice underwent the myocardial IR procedure and were randomized to receive i.v. metoprolol or vehicle. Capillary

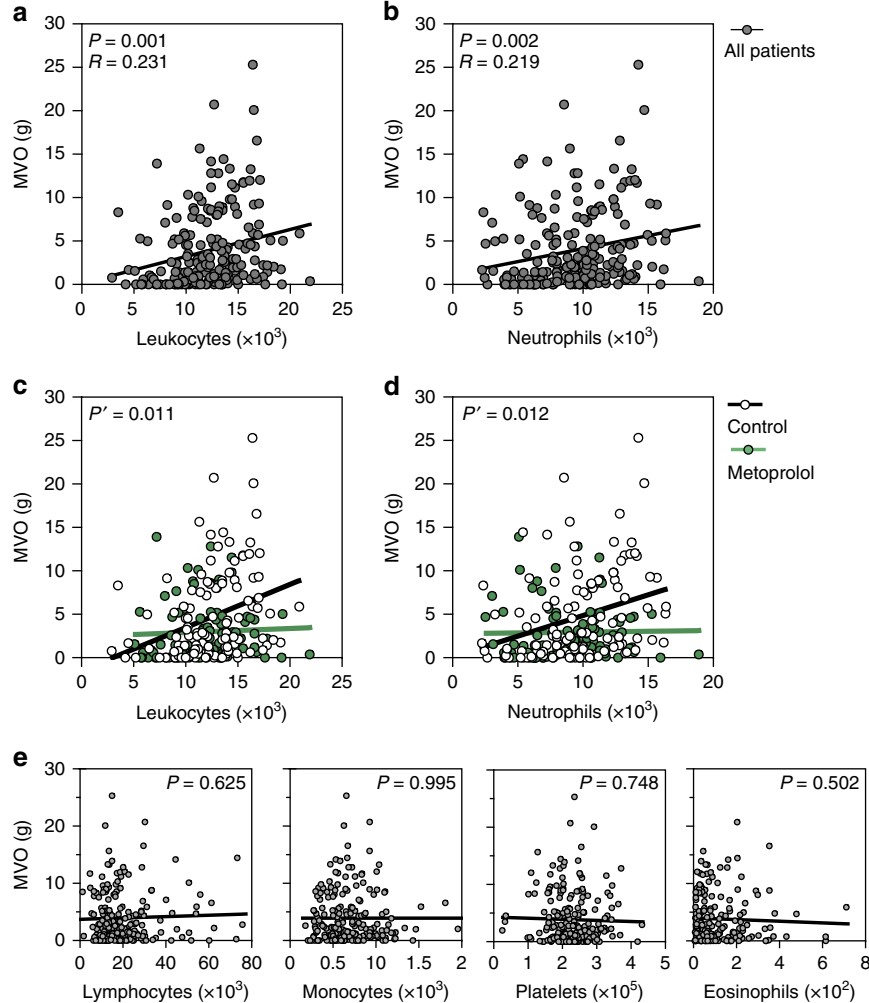

**Figure 2 | Metoprolol abrogates neutrophil count positive association with extent of MVO.** Sensitivity analysis of the association between MVO and leukocyte and subpopulations count on admission from METOCARD-CNIC trial patients. (**a**,**b**) Association between MVO and absolute leukocyte or neutrophil count on admission. Grey dots represent individual values and line linear relationship. (**c**,**d**) Linear regression comparison between MVO and leukocytes and neutrophils in the subsets of METOCARD patients indicating loss of correlation in the metoprolol treated group (green) as compared to control group (white). P', stands for interaction P value. (**e**) Association between MVO and rest of white blood cells subpopulations: Platelets, lymphocytes, eosinophils and monocytes showing no correlation in the extent of MVO. P stands for P value and R, for Pearson's correlation coefficient.

obliteration (a histological surrogate for MVO) and leukocyte infiltration were quantified at 6 h post-reperfusion (Fig. 3e–m). Metoprolol administration during ongoing AMI resulted in a significant reduction of capillary obliteration by circulatory cell plugins when evaluated at 6 h post-reperfusion (Fig. 3e,f). Confocal microscopy analyses revealed a significant reduction in the number of myeloid cells plugins (LysM-GFP+ particles) and percentage of LysM-GFP+ area within the LV sections, indicating rapid inhibition of leukocyte recruitment and protection of lumen vessel patency (Fig. 3g, Supplementary Fig. 2a–d and Supplementary Movie 1). Temporal (6 and 24 h) evaluation of myeloid-derived cells infiltration into injured myocardium showed a persistent abrogation of neutrophil infiltration (Fig. 3h–m and Supplementary Movie 2) and a differential relative proportion of different myeloid cells in hearts from metoprolol and vehicle-treated mice (Fig. 3j and Supplementary Fig. 3). These mouse experiments confirm the clinical findings that pre-reperfusion metoprolol administration during ongoing AMI limits infarct size and reduces MVO, and further show that metoprolol reduces neutrophil infiltration, suggesting neutrophils as a potential target in this cardioprotective effect.

**Metoprolol does not protect from AMI in the absence of neutrophils.** Catecholamine-stimulation of βARs alters neutrophil function, cytokine release and neutrophil–platelet aggregate formation[17,23,24], processes associated with aggravated injury during AMI (refs 25–27). To decipher the role of neutrophils in the protection afforded by metoprolol during ongoing AMI, we evaluated the effect of the drug in the absence of neutrophils. Neutrophil depletion in mouse peripheral blood was achieved by administration of an anti-Ly6G mAb over 2 days[8,28]. After neutrophil depletion, animals were subjected to myocardial IR, and infarct size was evaluated at 24 h post-reperfusion (Fig. 3n). Confirming earlier reports[29], neutrophil-depleted mice had smaller infarcts than controls. Administration of metoprolol to these mice during ongoing AMI did not reduce infarct size (Fig. 3o–q). The abrogation of the cardioprotective effect of metoprolol confirms circulating neutrophils as a target of the beneficial effect associated with this pharmacological therapy.

**Metoprolol inhibits neutrophil migration by targeting ADRB1.** The effect of metoprolol on primary neutrophils' function was

evaluated in a chemokine-induced transwell migration assay and by evaluation of the chemotactic FPR activator-peptide, W-peptide[30], -induced reactive oxygen species (ROS) production assay[31]. First, neutrophils were exposed across the transwell filter to the chemoattractant CXCL1 in the presence or absence of metoprolol for 1.5 h, and migration through the transwell membrane was quantified by flow cytometry. Metoprolol inhibited baseline and epinephrine-stimulated neutrophil migration towards CXCL1, reducing migration to the same level in both cases (Fig. 4a). Second, ROS production *in vitro* was measured using DHR 123 and W-peptide activation. ROS production was tested in metoprolol-treated and untreated neutrophils with and without W-peptide stimulation.

Metoprolol-treated neutrophils presented significant decreased oxidative burst compared to the non-treated cells after stimulation (Fig. 4b,c). Metoprolol alone had no effect on ROS endogenous production.

Metoprolol is a selective ADRB1-blocker, and ADRB1 signalling has been shown to mediate some of the pro-inflammatory response of monocytes[32]. Neutrophils and monocytes are both myeloid derived cells, and we therefore reasoned that ADRB1 might be involved in the anti-migratory effect of metoprolol. *Adrb1* mRNA expression in fresh and viable isolated neutrophils was confirmed by PCR in wild-type mice and absence in neutrophils from *Adrb1*-knockout mice (*Adrb1*KO) (Supplementary Fig. 4a–c). *In vitro*, the -epinephrine-mediated

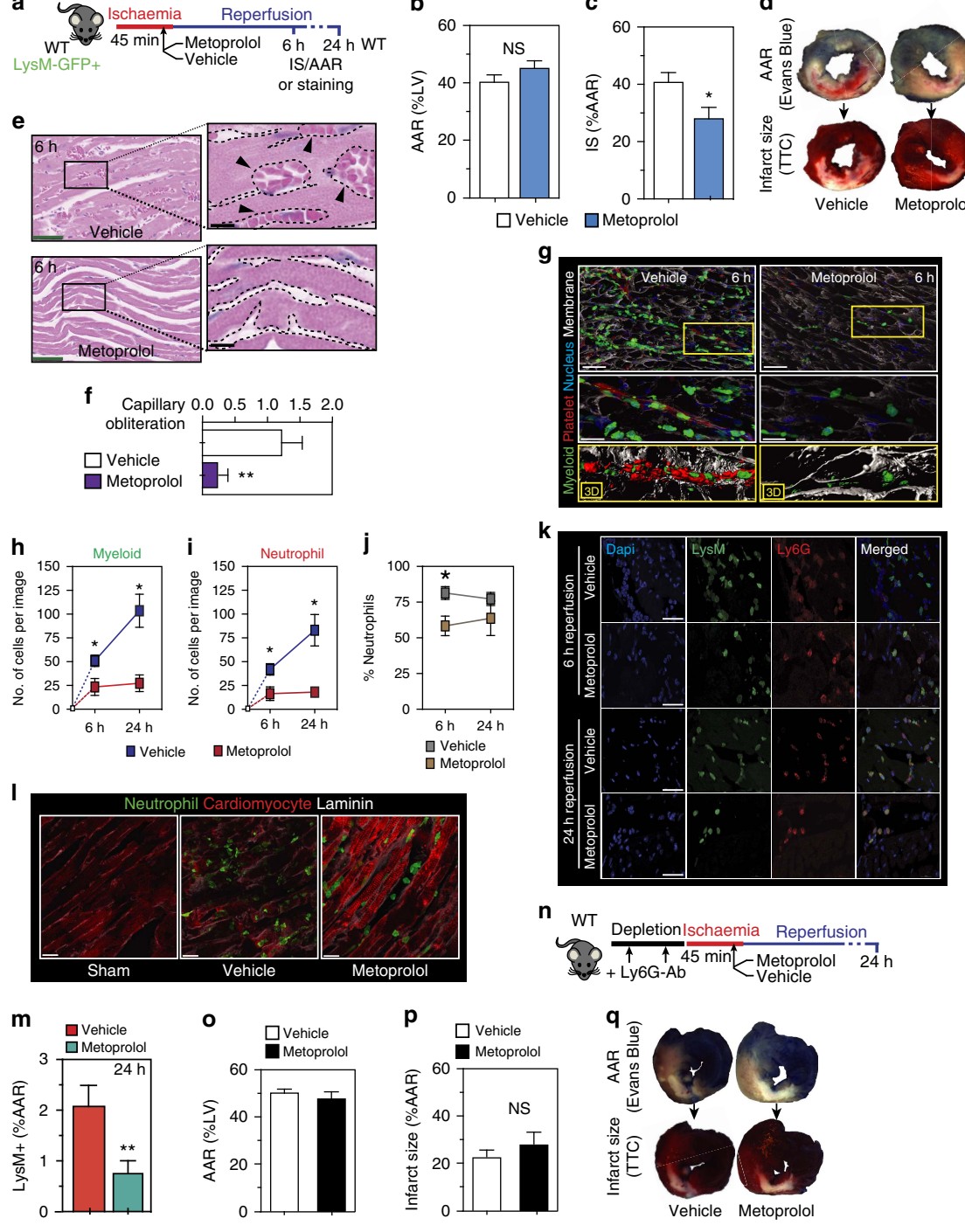

migration- and -W-peptide-mediated ROS production-inhibitory effect of metoprolol was lost in *Adrb1*KO neutrophils (Fig. 4a–c). β2-adrenergic receptor-knockout mice (*Adrb2*KO) was not involved in the effects observed after metoprolol administration (Supplementary Fig. 5).

We next explored whether metoprolol directly inhibits the capacity of neutrophils to infiltrate tissues *in vivo*. For this, we first used a model of thioglycolate-induced peritonitis (Fig. 4d). Thioglycolate induces massive leucocyte migration into the peritoneal cavity within the first 16 h (Supplementary Fig. 6a), with the majority of infiltrating cells being neutrophils (Supplementary Fig. 6b–d). Metoprolol i.v. administration sharply inhibited thioglycolate-induced neutrophil infiltration into the peritoneal cavity of wild-type mice (Fig. 4e,f), but the inhibitory effect of metoprolol was lost in *Adrb1*KO mice (Fig. 4f and Supplementary Fig. 6a,b).

To better define the cell compartment targeted by metoprolol, we lethally irradiated *Adrb1*KO mice and restored haematopoiesis with bone marrow (BM) transplanted from wild-type donors, generating chimeric mice expressing ADRB1 only in circulating cells. At 4 weeks after irradiation, transplanted animals presented >85% BM engraftment (Supplementary Fig. 7) and were subjected to thioglycolate-induced peritonitis. The replenishment of ADRB1 only in haematopoietic cells was enough to rescue the anti-leucocyte-infiltration effect of metoprolol (Fig. 4g). Together, these data show a direct effect of metoprolol on neutrophil function and demonstrate that the presence of ADRB1 in circulating cells is essential for the ability of metoprolol to reduce neutrophil infiltration into injured tissue.

**IR reduction by metoprolol involves haematopoietic cells' ADRB1.** We next investigated the involvement of ADRB1 blockade in haematopoietic circulating cells in the protective effect of metoprolol in the infarcted myocardium. *Adrb1*KO mice were subjected to myocardial IR and randomized to receive i.v. metoprolol or vehicle during ongoing AMI (Fig. 4h). Unlike the situation in wild-type mice, metoprolol did not limit infarct size in *Adrb1*KO animals (Fig. 4i–k), demonstrating the critical role of ADRB1 blockade in the cardioprotective effect. To demonstrate the role of ADRB1 expression in haematopoietic circulating cells, we repeated the myocardial IR protocol in the chimeras described above (*Adrb1*KO mice transplanted with wild-type BM). The presence of ADRB1 only in haematopoietic circulating cells was sufficient to restore susceptibility to the cardioprotective effect of metoprolol (Fig. 4l–n). Conversely, transplant of *Adrb1*KO

BM into irradiated wild-type mice abrogated the protective phenotype associated with metoprolol administration during IR (Fig. 4o–q). These data confirm the involvement of ADRB1 blockade in haematopoietic cells in the cardioprotection afforded by metoprolol administration during AMI.

**Metoprolol alters neutrophil dynamics *in vivo*.** During acute injury, neutrophils alter their morphology upon adhering to the activated endothelium. These shape change or polarization, permit intercellular interactions critical for the inflammatory response in several conditions, including myocardial IR (refs 8,33). Polarization of adhered neutrophils involves receptor redistribution and the assembly of a rearward protruding microdomain called the uropod, and is essential for the integration of signals coming from the endothelium and activated platelets before infiltration[8]. We were therefore interested in investigating whether metoprolol impaired neutrophil migration and infiltration through an effect on neutrophil dynamics. For this, we used bi-dimensional and 3D intravital microscopy (IVM) to image cremaster muscle vessels of mice treated with tumour necrosis factor-α (TNFα), an inflammatory model in which the vast majority of recruited leukocytes are neutrophils. Neutrophil behaviour was evaluated 3 h after administration of metoprolol or vehicle (Fig. 5a). Notably, metoprolol reduced neutrophil migratory velocity, accumulated crawling distance and directional movement (Fig. 5b,c and online Supplementary Movie 3). Independent-neutrophil 3D-reconstructions of live inflamed vessels showed that metoprolol consistently disabled the intravascular behaviour of neutrophils without disrupting polarization within activated vessels, resulting in dramatic changes in cell morphology (Fig. 5d–f) that correlated with their abnormal crawling dynamics. These data show that metoprolol 'stuns' neutrophils, resulting in altered dynamics and prevents the morphological changes needed to initiate intercellular interactions and subsequent tissue infiltration.

**Metoprolol blocks PSGL1-dependent neutrophil platelet interactions.** Correct neutrophil polarization and organization of an extruding microdomain that captures circulating platelets is required to initiate tissue-damaging inflammation[8]. Indeed, plugs of neutrophil–platelet co-aggregates in the microcirculation are a major contributor to MVO in AMI and in other models of injury[1,8]. We therefore explored the impact of metoprolol-induced neutrophil stunning on neutrophil–platelet interactions.

**Figure 3 | Metoprolol reduces neutrophil infiltration and capillary obliteration in murine IR.** (**a**) Mouse model of myocardial IR. (**b, c**) Histological evaluation of left ventricle (LV) area at risk (AAR) and infarct size (IS) in mice subjected to IR and randomized to receive metoprolol (blue) or vehicle (white); NS stands for non-significant. n = 8. (**d**) Representative images of LV slices showing AAR (negative for Evans Blue) in upper panels and extent of necrosis (triphenyltetrazoliumchloride (TTC)-negative area in lower panels). (**e**) Capillary oblliteration quantified in H&E ten random images; n = 5–6. (**f**) Representative H&E myocardial images at 6 h reperfusion showing disarrayed and abundant obstructed capillaries in the vehicle-treated sample; metoprolol-treated samples show injury and nuclear condensation but no signs of MVO; green scale bars, 50 μm. Detailed amplification of the boxes show obstructed capillaries indicated with black arrows; black scale bars, 10 μm. (**g**) Confocal images from LV at 6 h after reperfusion onset showing massive vascular neutrophil migration (LysM-GFP, green) and co-aggregates with platelets (CD41, red) vehicle- but not in metoprolol-treated mice; scale bar, 25 μm. Next, amplified boxes indicating regions of capillary obstruction; scale bar, 10 μm. Bottom yellow panels show computed 3D reconstructions. (**h,i**) Myeloid-derived cell infiltration dynamics showing maintained attenuation in hearts from metoprolol-treated mice. n = 5. (**j**) Neutrophilic proportions infiltrate dynamics. (**k**) Representative confocal images of LV sections taken from injured mice after 6 and 24 h reperfusion onset. Myeloid infiltration (LysM-GFP +, green) most of which are neutrophil (Ly6G +, red) is evident in vehicle-treated mice and significantly attenuated in those from metoprolol-treated mice; merged images show double positive cells (LysM +/Ly6G +, that is, neutrophils). Scale bar, 50 μm; n = 3–5. (**l**) Representative confocal images of neutrophil infiltration 24 h after reperfusion onset. Vehicle-treated mice show massive myocardial neutrophil infiltration (LysM-GFP +, green), with dispersed cells attached to the injured cardiac fibre membranes (α-actinin, red; laminin, grey). (**m**) LysM + total area in the LV section as a %AAR; scale bar, 20 μm; n = 5–6. (**n-q**) Effect of metoprolol on limiting-infarct size in neutrophil-depleted mice. (**n**) Neutrophil depletion model. (**o**) Myocardial area at risk (AAR). (**p**) Infarct size. (**q**) Representative transverse ventricular slices showing AAR and infarct size Data are means ± s.e.m. *P<0.05; **P<0.01, determined by the nonparametric Wilcoxon–Mann–Whitney test for each panel.

Using the cremaster IVM model of TNFα-induced local inflammation, we evaluated the acute neutrophil–platelet inhibitory effect of metoprolol in polarized neutrophils (Fig. 5). Metoprolol i.v. administration effectively inhibited interactions with the uropod, but not the leading edge (Fig. 5g,h) and rapidly reduced total neutrophil–platelet interactions (Fig. 5i–k). IVM experiments in *Adrb*1KO mice revealed no differences

between metoprolol-treated and vehicle-treated mice, implicating ADRB1 in the inhibitory effect of metoprolol on neutrophil–platelet interactions (Fig. 5j).

Based on these findings, we hypothesized that inhibition of neutrophil–platelet interactions underlies the inhibitory effect of metoprolol on MVO after myocardial IR. To test this, we first evaluated the effect of metoprolol on neutrophil–platelet

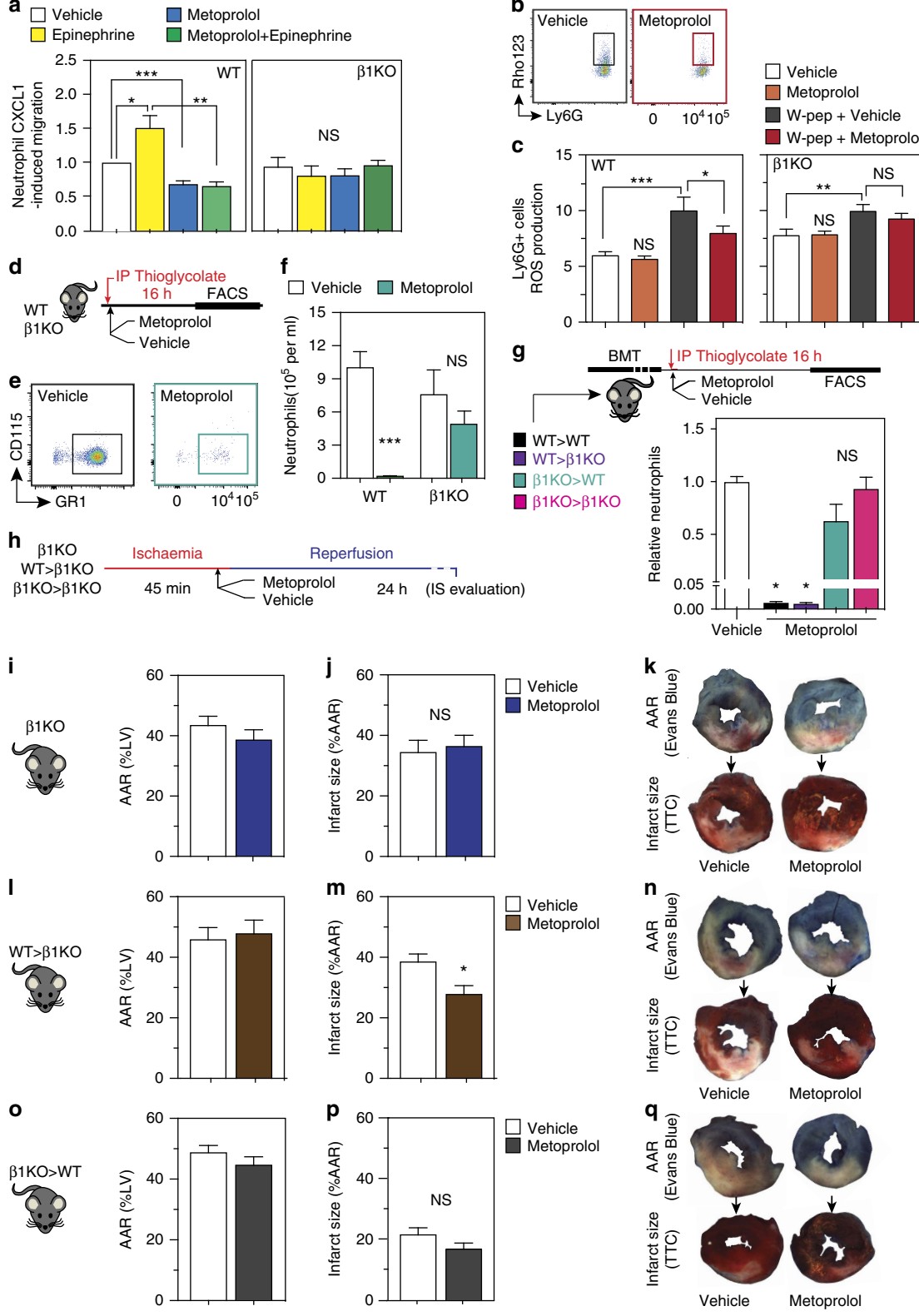

co-aggregate formation in mouse myocardial vessels after experimental IR. Administration of metoprolol to wild-type mice during ongoing AMI significantly reduced the number of neutrophils attached to the vessel wall (Fig. 6a,b) and the average number of interacting platelets per neutrophil (Fig. 6c).

Neutrophil–platelet interactions during acute injury are mediated by PSGL1 and signals delivered upon these contacts promote subsequent neutrophil extravasation and injury to the tissue[8]. Blockade of PSGL1 by pretreatment with PSGL1-mAB (Fig. 6e) significantly reduced infarct size in the myocardial IR model, and administration of metoprolol during ongoing AMI did not yield any further infarct-size reductions (Fig. 6f,g). These data confirm that metoprolol protects the infarcted myocardium by uncoupling neutrophil recruitment and polarization, thereby disrupting neutrophil–platelet interactions and the downstream inflammatory response.

**Metoprolol limits neutrophil–platelet aggregates in patients.** To investigate whether metoprolol alters neutrophil dynamics and inhibits neutrophil–platelet interactions in humans, whole blood drawn from healthy donors was incubated *ex vivo* with epinephrine (5 μM) and increasing concentrations of metoprolol (0, 2, 5 μM). Samples were then stained, and neutrophil–platelet co-aggregate formation was evaluated by flow cytometry[34–37] (morphological parameters, CD14neg, CD45 + , CD61 + ). Metoprolol significantly inhibited epinephrine-stimulated neutrophil–platelet co-aggregate formation (Fig. 7a). The effect of metoprolol *in vivo* was additionally studied in patients undergoing elective coronary angioplasty for acute coronary syndrome (ACS). Samples were collected before and after i.v. administration of metoprolol (15 mg) and circulating neutrophil–platelet co-aggregates were assessed by flow cytometry. Metoprolol administration significantly reduced the number of neutrophil–platelet interactions (Fig. 7b). To elucidate whether metoprolol was acting at the platelet level of action we evaluated effect of metoprolol on platelet function through platelet aggregation using light transmittance aggregometry in platelet-rich plasma (PRP) (Supplementary Fig. 8a) or platelet activation surface markers expression assay using flow cytometry (Supplementary Fig. 8b,c). Metoprolol did not show any effect on platelet aggregation/activation, which together with the aforementioned effects on neutrophil migration and ROS-production suggests that the effect seen on neutrophil–platelet co-aggregates was driven by a direct effect on neutrophils.

## Discussion

Early metoprolol administration during AMI, given as an adjunct to mechanical reperfusion has been shown to reduce infarct size and ameliorate post-infarction severe cardiac dysfunction[11,38].

Defining the mechanisms underlying this cardioprotection is therefore of great medical relevance since more efficient and specific protective strategies could be identified.

The ability of ADRB1 selective blockers to reduce infarct size was tested decades ago in several clinical trials, but the results were inconclusive[1]. However, most of these early studies were performed in the context of non-reperfused AMI. The advent of reperfusion as the treatment of choice for AMI changed the mode of myocardial death: from unrelieved ischaemia to a combination of ischaemia and reperfusion processes. Therefore the potential cardioprotective effects of ADRB1 selective blockers needed to be revisited in light of the current evidence of IR injury during AMI (ref. 1). The ability of metoprolol to reduce infarct size in the context of reperfused AMI was recently evaluated in the METOCARD-CNIC trial. In this trial, early i.v. metoprolol administration during ongoing AMI resulted in a significant reduction of infarct size[9,39], and also significantly reduced the incidence of severe ventricular dysfunction and heart failure readmissions[10]. Large animal studies conducted before the METOCARD-CNIC trial showed that metoprolol reduces infarct size only when administered before reperfusion[40,41], suggesting that metoprolol might reduce infarct size by inhibiting reperfusion injury. The early studies, testing the infarct-limiting properties of metoprolol in non-reperfused AMI, were undertaken under the hypothesis that ADRB1-blockers would reduce the extent of damage by a direct effect on cardiomyocytes via reducing oxygen consumption. Although our large animal study[40] did not investigate the mechanism of metoprolol-mediated protection, it did show that oxygen consumption was similar in metoprolol- and vehicle-treated pigs. In addition, the infarct-size reduction in metoprolol-treated pigs was associated with reduced myeloperoxidase activity in the post-ischaemic myocardium 24 h after reperfusion[41]. These findings challenged the idea that metoprolol could reduce cardiac damages simply by reducing myocardial oxygen consumption directly in the heart and prompted us to investigate the cellular mechanisms underlying the reperfusion-related injury-limiting effect of metoprolol during IR.

There is ample evidence supporting the critical implication of neutrophils in IR injury[4,25,27,29,42,43]. Neutrophils have two deleterious effects in this context. First, neutrophils and neutrophil–platelet plugs occlude microvessels, preventing efficient tissue perfusion: MVO. Second, neutrophils adhering to the newly reperfused injured vessels infiltrate the myocardium[1], prompting deleterious processes associated with reperfusion[25,33]. Our study confirms these associations in AMI patients: neutrophil count positively correlated with the extent of MVO as evaluated by state-of-the-art MRI 1 week after reperfusion. There exist weak pre-existing evidence linking the action of metoprolol to neutrophils, including reductions in

**Figure 4 | Metoprolol directly inhibits neutrophil deleterious function through a *ADRB1*-blockade. (a)** Effect of metoprolol on CXCL1-induced migration of fresh isolated primary neutrophils (Ly6G + ) from WT or *Adrb1*-knockout (β1KO) mice. CXCL1-stimulated cells were incubated with vehicle, epinephrine (10 μM), metoprolol (10 μM) and epinephrine + metoprolol; n = 4 independent experiments. NS, stands for non-significant. **(b,c)** Inhibitory effect of metoprolol on W-peptide-induced ROS production on fresh isolated primary neutrophils (Ly6G + ) from WT or β1KO mice. Mean fluorescent intensity of Rho123 in Ly6G + neutrophils after W-peptide stimulation. n = 6 independent experiments; flow cytometry plots illustrate reduced expression of Rho123 in metoprolol-treated neutrophils. **(d–f)** Effect of metoprolol on limiting-thioglycolate-induced peritoneal infiltration on WT and β1KO mice. **(e)** Flow cytometry plots illustrating reduced infiltration of neutrophils (CD115neg; GR1 + ) in metoprolol-treated mice. Absolute neutrophils detected per ml of infiltrate at 16 h after thioglycolate injection in WT mice (n = 7–9) or β1KO mice (n = 5). **(g)** Effect of metoprolol on thioglycolate-induced neutrophil infiltration in the four BMT groups. Protocol scheme for thioglycolate-induced peritonitis assay after bone-marrow transplants (BMT) between WT and β1KO mice, evaluating the influence of the presence or absence of *ADRB1* in circulating cells. Data are normalized to vehicle; n = 4–9. **(h)** Protocol scheme for IR experiments in chimeric animals after BMT, evaluating the infarct-limiting effect of metoprolol in the presence or absence of *ADRB1* in circulating cells. **(i–q)** Area at risk (AAR) and infarct size, as well as representative images of Evans blue and TTC staining in metoprolol-treated and vehicle-treated β1KO and chimeras (WT bone marrow transplanted into β1KO mice and reverse transplants). Infarct size is reduced by metoprolol only when circulating cells express β1-adrenergic receptor; n = 9 (**i,j**), n = 8 (**l,m**), n = 5–6 (**o,p**). Data are means ± s.e.m. *P < 0.05; **P < 0.01; ***P < 0.001, determined by the nonparametric Wilcoxon–Mann–Whitney test or using the one-way ANOVA and Holm Sidak's *post-hoc* multiple comparisons method.

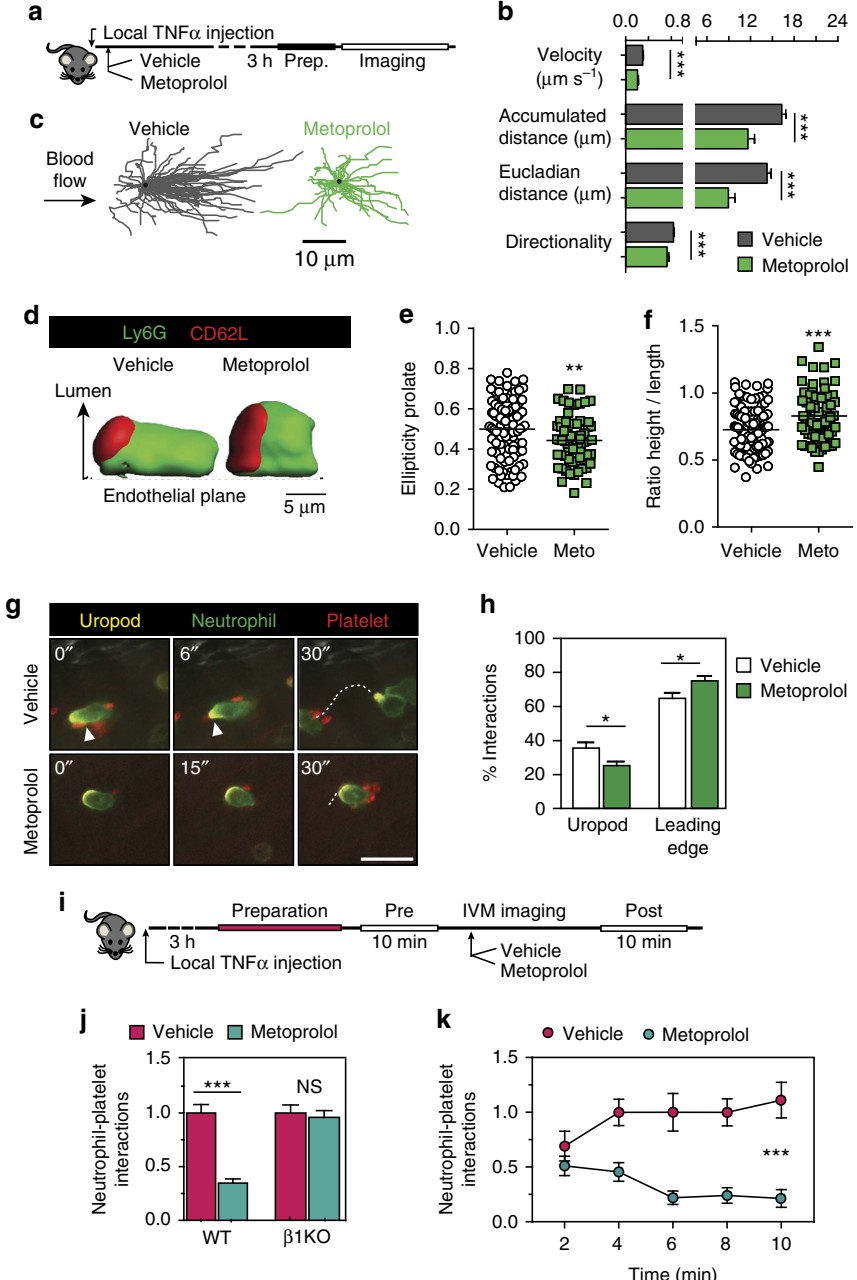

**Figure 5 | Metoprolol stuns neutrophils and prevents interactions with platelets.** (**a**) Experimental design: WT mice receiving TNFα were randomized to receive i.v. metoprolol or vehicle before analysis of cremaster muscle vessels by 2D and 3D intravital microscopy. (**b**) Quantification of parameters related to two-dimensional intravascular motility; $n = 54$–141 cells from 3 to 4 mice. (**c**) Representative tracks of crawling neutrophils within inflamed vessels. (**d**) 3D reconstructions of representative neutrophils within live vessels of saline-treated and metoprolol-treated mice (red, uropod; green, cell body). (**e,f**) Quantification of 3D parameters, indicating reduced elongation (prolate ellipticity) and enhanced projection of recruited neutrophils into the luminal space (height-to-length ratio); $n = 68$–105 cells from 3 to 4 mice. (**g,h**) Representative time-lapse images and percentage of interactions of platelets (CD41, red) with the polarized neutrophil uropod (CD62L, yellow) or leading edge (Ly6G, green); $n = 28$–29 vessels from 3 to 4 mice. White arrowheads indicate interactions with the uropod and the dotted line the displacement of the cells during 30 s. Scale bar, 10 μm. (**i**) Protocol scheme for evaluating the acute inhibitory effect of metoprolol on activated and polarized neutrophils on WT and *AdrbB1*-knockout (β1KO) mice. (**j**) Absolute neutrophil–platelet intravascular interactions in WT and β1KO mice. NS, stands for non-significant. (**k**) Temporal neutrophil–platelet interaction inhibition in WT mice after administration with i.v. metoprolol. Data are means ± s.e.m. *$P < 0.05$; **$P < 0.01$; ***$P < 0.001$, determined by unpaired Student's *t*-test for each parameter.

post-IR myeloperoxidase activity in pig myocardium[41] and rat spinal cord[44], and inhibition of sepsis-induced inflammation in mice[45]. The data here presented from patients randomized in a controlled clinical trial are the first human evidence linking metoprolol with altered neutrophil behaviour *in vivo*. We show that the strong positive correlation between neutrophil count and

MVO was abolished in patients receiving i.v. metoprolol before reperfusion. The fact that metoprolol-treated patients' high neutrophil count was not associated with the extent of MVO suggests an altered neutrophil dynamics during acute injury.

Some preclinical studies have suggested an association between metoprolol exposure and altered neutrophil dynamics[41,44,45];

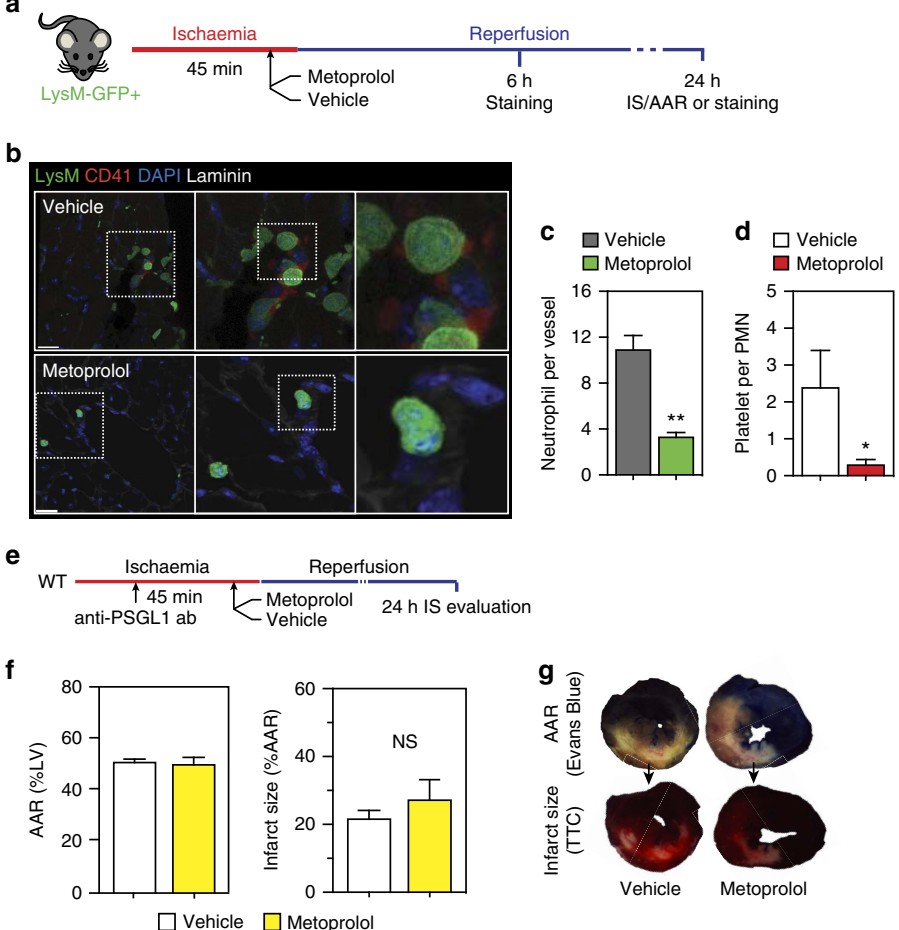

**Figure 6 | Metoprolol inhibits neutrophil–platelet interactions.** (**a**) Experimental scheme and representative confocal images evaluating the effect of metoprolol on the number of co-aggregates of neutrophils (LysM-GFP+, green) and platelets (CD41, red) in the post-reperfused mouse myocardium. (**b,c**) LysM-GFP+ cells (neutrophils) attached to coronary vessels (n = 7–9) and the numbers of interacting platelets (CD41+) per neutrophil in reperfused myocardium. Scale bar, 20 μm. (**d**) Protocol scheme for the IR experiment evaluating the effect of neutrophil-platelet blockade with anti-*PSGL1* Ab (administered 15 min after ischaemia onset, that is, 30 min before reperfusion) on the infarct-limiting effect of metoprolol (administered 35 min after ischaemia onset, that is,10 min before reperfusion). (**e-g**) AAR, infarct size, and representative images of Evans blue and TTC staining in vehicle- and metoprolol-treated mice pretreated with anti-*PSGL1* Ab. n = 5–7. NS, stands for non-significant. Data are means ± s.e.m. *P < 0.05; **P < 0.01. Comparison was determined by the nonparametric Wilcoxon–Mann–Whitney test.

however, the cellular mechanism responsible for this effect has remained unknown. Our results from the *in vitro* transwell migration assays confirm previous studies showing the ability of metoprolol to inhibit migration of isolated neutrophils[46]. Also, our *in vitro* data confirm previous studies showing that metoprolol was able to inhibit ROS production from neutrophils[47]. Our *in vitro* results showing migration inhibition by metoprolol are reinforced by the acute *in vivo* murine peritonitis model, in which a single metoprolol i.v. injection abrogates neutrophil infiltration into the peritoneal cavity. Furthermore, the studies with *Adrb1*KO mice provide evidence for a critical involvement of the ADRB1 axis in the effects of metoprolol. In the transwell assays, metoprolol had no effect on the migration of *Adrb1*KO neutrophils, and in the peritonitis assay, the inhibitory effect of metoprolol on neutrophil infiltration was lost in *Adrb1*KO mice. The rescue of this inhibitory effect in chimeric animals in which *Adrb1* is expressed only in circulating cells demonstrates that the cellular target of metoprolol action is of haematopoietic origin, and rules out an involvement of other compartments. Moreover, the findings in the peritonitis model were confirmed in the myocardial IR model,

with metoprolol significantly reducing neutrophil infiltration into the injured myocardium of wild-type but not *Adrb1*KO mice, and rescue of the protective effect in the *Adrb1*KO after wild-type BM transplant. Finally, we ruled out any involvement of ADRB2 on the effect exerted by metoprolol. Altogether, these results demonstrate the essential role of neutrophil ADRB1 blockade in the protective effect of metoprolol against myocardial IR injury. Our data are in discrepancy with a prior study suggesting that metoprolol abrogated neutrophil migration *in vitro* in an apparently ADRB1-independent manner[46]. In that study authors reported that metoprolol was able to inhibit neutrophil migration even in the presence of the β-adrenergic receptor agonist orciprenaline, while in our study we have used a genetic ablation toll (that is, *Adrb1*KO mice), which ensures the absence of ADRB1.

Upon acute injury, neutrophils recruited to injured vessels initiate inflammation by scanning for activated platelets present in the circulation, establishing interactions with them through *PSGL1* exposed on neutrophil protrusions[8]. The IVM confocal imaging analysis demonstrated that metoprolol prevents the exposure of functional *PSGL1* clusters, which are essential for

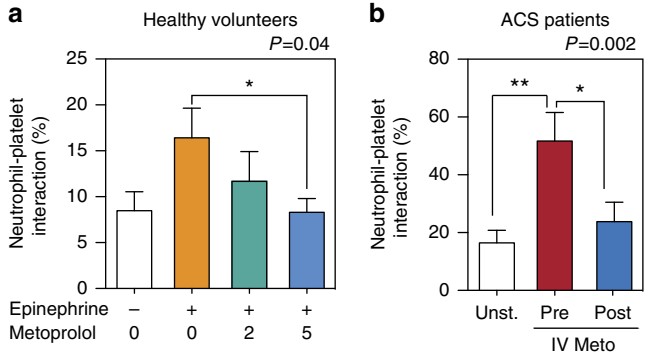

**Figure 7 | Metoprolol inhibits neutrophil–platelet interactions in patients.** (**a**) Effect of metoprolol on neutrophil–platelet co-aggregate formation in epinephrine-stimulated whole blood from healthy volunteers ($n = 20$). Whole blood was incubated *in vitro* with epinephrine 5 μM and metroprolol (Meto, concentrations in μM). (**b**) *In vivo* effect of metoprolol (up to15 mg i.v.) on the number of neutrophil i.v.platelet co-aggregates in acute coronary syndrome (ACS) patients scheduled for coronary angioplasty. Blood was drawn before and after metoprolol i.v. administration; $n = 6$ ACS patients. Pre, before i.v. administration; Post, after i.v. administration. Data are means ± s.e.m. *$P < 0.05$; **$P < 0.01$, determined by one-way ANOVA and Holm Sidak's *post-hoc* multiple comparisons method.

interaction with platelets. These events correlated with alterations in neutrophil properties essential for infiltration, including migratory velocity and directionality. In agreement with these data, in the mouse myocardial IR model, metoprolol altered neutrophil 3D structure and reduced the numbers of neutrophil–platelet co-aggregates that occluded myocardial vessels. The inability of metoprolol to provide additional protection after blockade of neutrophil–platelet interactions with the anti-PSGL1 mAB is compelling evidence for interference in this interaction as the mechanism underlying the infarct-limiting action of metoprolol during myocardial IR in mice. The demonstration that metoprolol inhibits neutrophil–platelet interactions in healthy volunteers and ACS patients suggests that this mechanism also operates in humans. There is controversy on the direct effect of metoprolol on platelet aggregation. Some studies suggested that metoprolol was able to inhibit ADP- and epinephrine-mediated platelet aggregation[48], but recent studies did not show any anti-platelet effect of metoprolol[49]. In line with the latter evidence, in our study we found that metoprolol had no effect on platelet aggregation or in platelet activation in healthy volunteers. These data strongly suggest that the inhibitory effect of metoprolol on neutrophil–platelet co-aggregates was driven by a direct effect on neutrophils. In our flow cytometry studies we defined a positive neutrophil–platelet interaction as CD14neg, CD45 +, CD61 + particles. It should be noted that there are more specific neutrophil markers and thus our selection is a limitation of these experiments. However, given that all data (human and mouse) point to the same direction, we think this limitation did not have an impact on the conclusions obtained in these experiments.

The role of neutrophils in experimental IR injury is well established; however, the negative results of clinical trials with anti-inflammatory therapies (for example, C5a (refs 50,51), CD-18 (ref. 52)) dampened hopes for this pharmacological strategy[3], and suggested that parallel mechanisms driving myocardial injury were at play. The demonstration that neutrophil dynamics in general and neutrophil–platelet interactions in particular are the target for the protective actions of metoprolol

administered to patients during AMI reinstates neutrophils as a potential target for the therapeutic reduction of infarct size.

## Methods

**Human cardiac magnetic resonance imaging.** The METOCARD-CNIC trial (NCT01311700) recruited patients suffering an AMI during hospital transit to undergo mechanical reperfusion by primary angioplasty. Patients were randomized to receive i.v. metoprolol (up to 15 mg) or no drug (control). Patients underwent two CMR studies: 1 week and 6 months after AMI. Images were acquired with a 3.0 Tesla magnet (Achieva Tx, Philips Medical Systems) with vectorcardiographic gating and a dedicated cardiac 32-channel phased-array surface coil. The extent of MVO was measured in the 1-week CMR study; to detect and quantify MVO, a delayed enhancement imaging was performed 10 min after gadolinium contrast injection, using a T1-weighted 2-D Inversion Recovery Turbo Field Echo (2D IR-TFE) sequence. Myocardial necrosis was defined by the extent of abnormal gadolinium enhancement, whereas MVO was defined as black-hypoenhanced areas within the bright-hyperenhanced regions. CMR analysis was undertaken by operators blinded to treatment allocation at the Centro Nacional de Investigaciones Cardiovasculares Carlos III (CNIC). Myocardial necrosis and MVO were quantified by semiautomatic delineation with dedicated software (QMass MR 7.6; Medis, Leiden, the Netherlands). Total MVO was quantified as grams of LV. To correct for infarct size, MVO was also expressed as a percentage of the infarcted area.

**Mouse procedures.** Experimental procedures were approved by the CNIC Animal Care and Ethics Committee and regional authorities. IR, thioglycolate-induced peritonitis and IVM experiments were performed in 8–13-week-old wild-type male C57BL/6 mice. β1-adrenergic receptor (*Adrb1*) knockout (KO; *Adrb1*KO) mice were in a mixed background. β2-adrenergic receptor (*Adrb2*) knockout (KO; *Adrb2*KO) mice were in a C5BL/6 background. For the BM transplant experiments, *Adrb1*KO mice were backcrossed with mice expressing DsRed under the control of the β-actin promoter to facilitate evaluation of BM engraftment. Male and female mice were used as donors in BM transplant procedures, but only males were used in myocardial IR and thioglycolate-induced peritonitis experiments. All animals were randomized to receive a single i.v. injection (50 μl) of metoprolol-tartrate (10 mM) or vehicle (saline). Histological evaluation of injured myocardium in the myocardial IR model was performed in lysozyme M-GFP + (LysM-GFP)[53] male mice. Intravascular neutrophil and neutrophil–platelet interactions were scored manually in the myocardium. Neutrophils were depleted in C57BL/6 male by i.v. injections of 50 μg anti-mouse Ly6G 24 and 48 h before the myocardial IR procedure[28]. For *in vivo* blocking of P-selectin glycoprotein ligand-1 (*PSGL1*), 50 μg of anti-*PSGL1* antibody (clone 4RA10) was i.v. injected 15 min after ischaemia onset. Mice were maintained under pathogen-free conditions in a temperature-controlled room and a 12-h light–dark cycle at the CNIC animal facilities. Chow and water were available *ad libitum*.

**Reagents.** Metoprolol-tartrate (M5391), Evans blue, triphenyltetrazolium chloride, DAPI (D8417-1MG), Mowiol mounting medium (81381), anti-laminin (L9393), anti-α-actinin (A7732), dihydrorhodamine 123 (D1054) were obtained from Sigma-Aldrich. Dylight-650-conjugated anti-1A8 Ly6G (BE0075-1) and anti-*PSGL1* antibody (clone 4RA10) from BioXcell. Anti-CD41 (12-04-11-83) and anti-CD115 (12-1152-83) were obtained from ebioscience. O.C.T. was obtained from Tissue-Tek. Qiagen RNeasy Plus Mini Kit (74136). Percoll Plus (17-5445-02) and Ready-to-go RT-PCR Beads (27-9259-01) from GE Healthcare. CXCL1 was obtained from (453-KC-010) from R&D Systems. W-peptide (WKYMVM, 1799) from Tocris. Thioglycolate (BD211716) and anti-GP IIb/IIIa from BD Biosciences. Anti-GR1 (ab2557) was obtained from Abcam and AF-647 (A-21472) from Molecular Probes. Anti-CD45-FITC from Miltenyi Biotec, Germany. PC5-conjugated anti-CD14, anti-CD61 PC7 and Versalyse solution were obtained from Beckman Coulter.

**Mouse model of myocardial IR injury.** Male 8–12-week-old mice were subjected to 45 min occlusion of the LAD coronary artery followed by 6 or 24 h of reperfusion. For infarct size evaluation, reperfusion was maintained for 24 h. For analysis of MVO, neutrophil infiltration and neutrophil–platelet interactions, reperfusion was maintained for 6 or 24 h as indicated. The IR procedure was performed as previously described[54]. Briefly, fully asleep animals were intubated and temperature controlled throughout the experiment at 36.5 °C to prevent hypothermic cardioprotection. Thoracotomy was then performed and the LAD was ligated with a nylon 8/0 monofilament suture for 45 min. The electrocardiogram was monitored (MP36R, Biopac Systems Inc.) to confirm total coronary artery occlusion (ST-segment elevation) throughout the 45 min ischaemia. Ten minutes before reperfusion onset, mice were randomized to receive a single i.v. injection (50 μl) of metoprolol-tartrate (10 mM) or vehicle (saline) through the femoral vein. Dose of metoprolol was chosen after a dose response study (Supplementary Fig. 9). At the end of reperfusion, the chest was closed and animals were kept with 100% O$_2$ and analgesized with buprenorphine (S.C., 0.1 mg per kg) until the end of reperfusion.

**Mouse infarct size quantification.** At the end of follow up, mice were re-anesthetized and re-intubated, and the LAD coronary artery was re-occluded by ligating the suture in the same position as the original infarction[54]. Animals were then killed and rapidly 1 ml of 1% (w per v) Evans Blue dye was infused i.v. to delineate AAR: myocardium lacking blood flow, that is, negative for blue dye staining. The heart was than harvested, LV was isolated, cut into transverse slices (5–7 1-mm thick slices per LV) and both sides were imaged. Sections post-Evans blue staining present two different areas: one palish negative for Evans blue perfusion, delineating AAR, and another blueish (positive Evans blue) area indicating remote tissue. In order to differentiate infarcted from viable tissue, same slices were incubated in triphenyltetrazolium chloride (TTC, 1% (w per v) diluted in PBS) at 37 °C for 15 min in constant shacking. The slices were then re-photographed and weighed. Post TTC incubation, Evans blue staining clears out and slices present two areas: one necrotic (palish negative to TTC staining) and one reddish alive (positive to TTC staining). Regions negative for Evans Blue staining (AAR) and for TTC (infarcted myocardium) were quantified using ImageJ (NIH, Bethesda, MD, USA) by blinded observer. Percentage values for AAR and infarcted myocardium were corrected to mg independently for each slice. Absolute AAR and infarct size were determined as the mg:mg ratio of AAR:LV and infarcted myocardium:AAR, respectively. Animals exceeding 80% of IS were excluded assuming absence of reperfusion.

**Hearts processing for histology.** At the end of reperfusion, LysM-GFP mice were re-anesthetized, placed in a supine position, their ventral thoracic regions wiped with 70% alcohol and killed by cervical dislocation. Next, 10 ml of PBS with heparin (50 U per ml) was gently infused through the vena cava to avoid blood clots. Hearts were then removed and cut into 1 mm-thick transverse sections and fixed with 2% PFA in PBS for 24 h at 4 °C.

Heart slices (1 mm) designated to histopathological analysis of capillary obliteration were dehydrated through an ethanol series, cleared in xylene, embedded in paraffin wax and consequently sectioned (4 μm) for staining with haematoxylin and eosin. All immunohistochemical procedures were performed with an automated autostainer (Autostainer Plus, Dako) at the CNIC Histology Unit. Images were digitally scanned (Nanozoomer-RS C110730, Hamamatsu) and examined with image analysis software (Tissuemorph, Visiopharm) by blinded observers. Once the lesion was identified, 10–12 images (×20) were taken at random, and capillary obliteration was scored from 0 to 2, with 0 indicating the absence of capillary obliteration and 2 indicating presence of obstruction in all capillaries.

**Confocal microscopy.** Sections designated for immunofluorescence staining and confocal microscopy were post-fixed overnight and placed in 30% sucrose for 24 h and included in O.C.T. (Tissue-Tek). Serial 4 μm coronal sections were cut on a freezing microtome (Leica CM1950) and stored in cryoprotective solution. Sections designated for evaluation of neutrophil infiltration of the injured area were washed in PBS for 15 min and incubated with DAPI (1:1,000) at room temperature for 5 min, washed in PBS twice and mounted in Mowiol mounting medium. Images of full short-axis heart sections were acquired with an inverted confocal laser imaging system (Zeiss LSM7004 4-Laser) and reconstructed with Zeiss ZEN reprocessing software. Presence of Lysm-GFP + cells analysis was performed with ImageJ (NIH) by blinded observers. Values of total GFP + surface were normalized to LV section surface. A second set of sections were stained with Dylight-650-conjugated anti-1A8 Ly6G and DAPI for quantification of specific neutrophil identification and temporal infiltration analysis. Eight independent 40 × were evaluated from injured myocardium from each animal, and presence of myeloid derived cells (LysM +), neutrophils (LysM + Ly6G +) and monocytes (LysM + Ly6Gneg) were identified and analysed using ImageJ by blinded observers. Four animals were stained for basal quantification. Some sections were stained with anti-laminin to stain cell membranes (1:150) and anti-α-actinin to visualize sarcomeres in cardiac fibres (1:200) for detailed illustration of neutrophil infiltration in the injured myocardium. Images were post-processed and edited using the Imaris software (Bitplane AG, Switzerland) as indicated below.

Neutrophil–platelet interactions staining in infarcted mouse heart was adapted from Sreeramkumar et al.[8]. Briefly, OCT-embedded heart slices were cut into 50 μm sections, washed in PBS for 15 min and incubated for 20 min at room temperature in blocking buffer (PBS containing 10% BSA and 2% goat serum). For detailed visualization and characterization of the injured zone, cell membranes were stained with anti-laminin (1:150) and platelets were stained with PE-conjugated anti-CD41 (1:200). Primary and secondary antibodies were diluted 1:200 in blocking buffer, and incubations were conducted for 1 h at room temperature. Nonspecific staining was assessed by omission of primary antibody. Samples were counterstained with DAPI and mounted in Mowiol. Images were acquired with a laser-scanning confocal imaging system (Leica SP8 or SP5) at the CNIC Microscopy Unit and post-processed with Leica Las AF and Imaris software (Bitplane AG, Switzerland). Independent vessels (7–9) were evaluated for each animal, and intravascular neutrophil and neutrophil–platelet interactions were scored manually.

**Neutrophil purification and *Adrb1* expression.** *Adrb1* expression in circulating neutrophils was examined in blood drawn from wild-type or *Adrb1*KO mice 20 min after injection of heparin (50 μl of 50 U per ml). Whole blood was filtered and pooled, and polymorphonuclear leukocytes were purified by gradient-centrifugation (800*g*, 20 min, 4 °C) in 65% Percoll Plus in Hanks balanced salt solution (HBSS). Cells were washed in PBS and residual erythrocytes were lysed using hypotonic buffer. Neutrophils were washed and resuspended in HBSS. Before RNA isolation, neutrophil identity was confirmed by flow cytometry (anti-1A8 Ly6G) and viability evaluated.

Total RNA from whole hearts, BM and purified neutrophils samples was isolated with the Qiagen RNeasy Plus Mini Kit. RNA (1–2 μg) was reverse transcribed using Ready-to-go RT–PCR Beads (27-9259-01, GE Healthcare). PCR was performed with 40 cycles of 95 °C for 12 s and 60 °C for 1 min. All PCR reactions were done in triplicate. Primers for Hprt and *Adrb1* were as follows: mHprt_fw-5′-GAGGAGTCCTGTTGATGTTGCCAG-3′, mHprt_rv-5′-GGCTGG CCTATAGGCTCATAGTGC-3′; m*Adrb1*_fw-5′-GTGGGTAACGTGCTG GTGAT-3′, m*Adrb1*_rv-5′-GAAGTCCAGAGCTCGCAGAA-3′. Amplicons generated in the qPCR were loaded on to an agarose gel to confirm single PCR products.

**Migration transwell assay.** The ability of leukocytes to migrate towards chemokine (C-X-C motif) ligand 1 (CXCL1) was assessed using a modification of the method of Villablanca et al.[55]. Briefly, wild-type or *Adrb1*KO mice were heparinized (50 μl of 50 U per ml, IP) and 20 min later blood was collected and filtered, and residual erythrocytes were lysed with hypotonic buffer. PBS-washed leukocytes of the same genotype were pooled and resuspended in RPMI containing 10% FBS and the appropriate treatment: saline (vehicle control), 10 μM epinephrine (positive control), 10 μM metoprolol-tartrate, or a combination of epinephrine and metoprolol. Transwell inserts (6.5 mm, 5.0 μm pore size (3421; Corning Costar Corporation)) were pretreated with 50 μl RPMI for 20 min and placed in 24-well-plates before seeding cells (100 μl; ~1 × 10⁵; >90% viability). Lower compartments (wells) were filled with 600 μl DMEM medium containing 0.04 ng per μl CXCL1 to induce directional movement. Spontaneous migration was assessed in wells lacking CXCL1. After incubation at 37 °C for 1.5 h, cells in the lower compartment were collected and neutrophils (Ly6G + cells) were evaluated by flow cytometry. Each independent experiment was conducted with leukocytes pooled from nine animals, and each of the five conditions was run with four replicates. Mean spontaneous migration was subtracted from the migration value of each well, and neutrophil migration was expressed as a percentage of the total number of neutrophils seeded in the upper chamber at the start of the experiment. For comparison between experiments and genotypes, migration was normalized to the mean control (vehicle) value.

**Neutrophil oxidative burst assay.** Blood from wild-type or *Adrb1*KO mice was collected in heparinized tubes and distributed in 100 μl aliquots, erythrocytes were lysed with hypotonic buffer. After centrifugation leucocytes were first washed and then re-suspended in high glucose phenol red free DMEM. Cells were then incubated for 50 min with or without metoprolol-tartrate 100 μM at 37 °C. As previously described[30,31], dihydrorhodamine 123 (DHR 123, 1 μM), which converts to the fluorescent product rhodamine 123 (Rho 123) upon oxidation was then added to the medium and cells were stimulated with the *chemotactic FPR activator-peptide,* w-peptide 1 μM (WKYMVM, 1799 Tocris). After 20 min incubation at 37 °C the reaction was stopped in ice and cells were washed in cold HBSS containing DAPI 0.1 μg per ml (D8417, Sigma). Mean fluorescent intensity for Rho 123 was evaluated for neutrophils (Ly6G + cells) alive (DAPIneg) by flow cytometry.

**Model of thioglycolate-induced peritonitis.** To assess the ability of metoprolol to inhibit neutrophil recruitment, we used a well-established thioglycolate-induced peritonitis model (see Fig. 3). Wild-type mice were intraperitoneally injected with 1 ml of thioglycolate and immediately randomized to receive a single 50 μl i.v. injection of vehicle or metoprolol-tartrate (10 mM). Sixteen hours later, 100 μl of blood from each animal was collected into EDTA tubes for later haematological analysis in a haematocytometer (Pentra 80). Next, animals were killed, 2 ml PBS was injected intraperitoneally and distributed manually for 30 s to detach infiltrated circulatory cells. Next, another 8 ml PBS was injected to facilitate collection of 6 ml peritoneal fluid. Exudates were gently centrifuged, and cells were washed twice with PBS and incubated for 1 h with anti-GR1 (1:200) and PE-conjugated anti-CD115 (1:200). After washing with PBS, cells were incubated for 30 min with anti-rat 647 to detect GR1. Cell nuclei were stained with DAPI. All samples were analysed by flow cytometry for exactly 30 s of constant flow. Neutrophil recruitment efficiencies are presented as neutrophils per ml of exudate for each independent animal.

To evaluate the role of ADRB1 in different compartments, we performed the same experiments in *Adrb1*KO mice and generated cohorts of chimeric mice by transplantation with BM cells from wild-type and *Adrb1*KO donors. Four weeks after BM transplantation, chimerism was evaluated by flow cytometry as the percentage of donated cells. Animals with chimerism below 85% were discarded; those with chimerism above 85% underwent the thioglycolate-induced peritonitis

protocol followed by randomization to receive either i.v. vehicle or metoprolol (10 mM). To compare between the different chimeric groups, the mean value for each metoprolol-treated group was normalized to the mean for the vehicle-treated group.

**Bone marrow transplant.** BM transplants protocols were adapted from Casanova et al.[28]. Recipient mice from *Adrb1*KO or wild-type genotypes (DsRed+ or DsRed- as appropriate) were lethally irradiated (13 Gy in two doses) before reconstitution with donor BM. Donor BM was collected from mice of the appropriate genotype by flushing both tibiae and femurs into PBS containing 2 mM EDTA (PEB buffer). Contaminating erythrocytes were lysed with hypotonic buffer. Engraftment in recipient animals was assessed by flow cytometry 3–4 weeks after transplantation. Animals bled for engraftment evaluations were rested for 1 week before any other procedure.

**Flow cytometry.** Neutrophil purity for *in vitro* migration assay and *Adrb1* expression analysis was evaluated by incubating cells with Dylight-650-conjugated anti-1A8 *Ly6G* and with DAPI to assess viability. Mouse primary blood leukocytes from peritonitis experiments were incubated with anti-Gr1 conjugated with AF-647 and with PE-conjugated anti-CD115 and DAPI. Neutrophils were gated on the basis of Gr1-positive and CD115-negative staining in a FACS Canto-3L flow cytometre equipped with DIVA software (BD Biosciences). Doublet discrimination and viability (negative to DAPI) was assessed for every sample. Data were analysed with FlowJo (Ashland) software by blinded observer. All experiments were conducted at the CNIC-Cellomics Unit.

**Intravital microscopy.** IVM of the cremaster muscle was performed after intrascrotal injection of TNFα (0.5 mg R&D Systems)[8], followed immediately by injection of a single i.v. bolus of metoprolol (10 mM) or vehicle, and neutrophil behaviour was evaluated 3 h after stimulus. In some experiments metoprolol was injected 3 h after treatment with TNF-α and images immediately acquired for analysis (see Fig. 4). The IVM system was built by 3i (Intelligent Imaging Innovations, Denver, CO, USA) on an Axio Examiner Z.1 workstation (Zeiss, Oberkochen, Germany) mounted on a 3-Dimensional Motorized Stage (Sutter Instrument, Novato, CA, USA). This set up allows precise computer-controlled lateral movement between XY positions and a Z focusing drive for confocal acquisition. The microscope is equipped with a CoolLED pE widefield fluorescence LED light source system (CoolLED Ltd. UK) and a quad pass filter cube with a Semrock Di01-R405/488/561/635 dichroic and a FF01-446/523/600/677 emitter. We used a plan-Apochromat ×40 W NA1.0 ∞/0 objective (Zeiss). Images were collected with a CoolSnap HQ2 camera (6.45 × 6.45-μm pixels, 1,392 × 1,040 pixel format; Photometrics, Tucson, AZ, USA). For confocal high-speed IVM, we used laser stacks for 488, 561 and 640 nm beams coupled with a confocal scanner (Yokogawa CSUX-A1; Yokogawa, Japan); images were acquired at 0.5 μm Z-intervals. Image acquisition was coordinated and offline data analysis facilitated with SlideBook software (Intelligent Imaging Innovations), run on a Dell Precision T7500 computer (Dell Inc., Round Rock, TX, USA). For three-dimensional analysis we used the 3D surface view function to determine the position of the CD62L+ clusters relative to the cell body and the lumen. Six to ten venules per mouse were analysed 210 to 300 min after TNF-α treatment by acquisition of fluorescence (Cy3/561 channels for phycoerythirn, FITC/488 channels for FITC and Cy5/640 channels for allophycocianin) and bright-field images with 2 × 2 for 2 min. For double staining with PE- and FITC-conjugated antibodies, acquisition was facilitated in single (FITC) and quad (PE) filters in order to avoid bleed-through of fluorescent signals between channels. For *in vivo* labelling of neutrophils and platelets surface molecules, fluorescently labelled antibodies were injected intravenously (anti-*CD62L*-FITC, anti-*Ly6G*-APC and anti-*CD41*-PE; 0.5–1.25 μg per mouse). Images were post-processed and edited using the Imaris software (Bitplane AG, Switzerland) as indicated below.

**Analysis of blood cell interactions.** Platelets in the inflamed cremaster muscle were visualized as CD41-labelled cells and quantified as reported[8]. Briefly, we defined the uropod of adherent neutrophils as the domain staining positive for CD62L, and the leading edge as the CD62L-negative pole forming multiple protrusions and showing guided movement. Six to ten venules per mouse were recorded, and platelet interactions with neutrophils were counted and analysed manually at the two distinct domains of the polarized neutrophil with the help of Slidebook software.

**Analysis of tracking of crawling neutrophils.** Time-lapse movies of crawling neutrophils were analysed with ImageJ, which includes the Manual Tracking and the Chemotaxis and Migration Tool plugins. For each movie we first adjusted channel intensities and converted them into RGB format. Movies were rotated so that the vessels and the blood flow were positioned horizontally and oriented left-right. When necessary, the Background subtraction and Image stabilization pre-stablished plugins were applied to eliminate noise and reduce tissue twitching, respectively. Both plugins were set up with xy calibration values, which depend on the camera and microscope parameters, to convert pixels into linear measures, as

well as the time interval value between movie frames (3 s). Each polarized neutrophil (identified by a clear polarized morphology or uropod staining) was tracked manually for 1 min using the Manual Tracking Plugin, which generated a data set with the respective xy track coordinates. We then used the Chemotaxis and Migration Tool to plot and the velocity (μm per s), accumulated distance (μm), euclidean distance (μm) and directionality of the tracks obtained. The Euclidean distance is the length of the straight-line segment connecting the initial and finishing points, whereas the accumulated distance is the total length of the path covered by the cell. Directionality measures how straight the cell track is, and is calculated as the ratio of euclidean distance to accumulated distance.

**Analysis of 3D reconstructions of polarized neutrophils.** We measured the 3D features of intravascular neutrophils using Imaris Software (Bitplane, Oxford, UK). From the parameters provided by the ImarisCell module, we selected prolate ellipticity by obtaining the lengths of the three semi-axes, which correspond with the Ellipsoid axis parameters. A prolate ellipsoid (cigar-like shape) is one for which the polar radius is greater than the equatorial radius. For 3D cell reconstructions, we used the ImarisCell module to define the cell body. We then segmented a region of interest to enclose an individual cell within this region, so that the subsequent reconstruction fitted the real cell structure. Afterwards, the respective source channel from which the cells had to be computed was selected. For reconstruction analysis, we chose the *Ly6G*-APC channel as it is a membrane-bound protein that yields a strong signal and allows a good rendering of the actual cell morphology. ImarisCell module determines the cell threshold by calculating voxel (3D pixel) intensities from the enclosed cell and comparing them with the background intensity in the enclosed sub-region. To obtain the height-to-length ratios, we visually established the cell orientation with respect to the vessel wall surface. Sections of each polarized neutrophil were analysed to manually measure the height of the cell and maximum length (from the top view) with respect to the vessel wall. For this purpose, we used Imaris Section View, which shows the coordinates in the three display areas (xy top view, zy lateral view and xz front view), and the Extended Crosshairs of the Section View, which in turn allows selection of the z-stack planes to visualize the entire cell (not just one plane or section) in the three views. A snapshot of these three views from a single cell was taken and imported into ImageJ, where height and maximum length were measured from the different views after setting the capture scale.

**Human blood sampling.** Functional tests were performed in blood samples from 20 volunteers (36 ± 6 years, 15 men). Exclusion criteria were as follows: any antiplatelet, anticoagulant or anti-inflammatory drug taken within the 2 previous weeks; abnormal platelet or leukocyte count; or any history of abnormal bleeding, thrombosis, or active inflammatory disease. Written consent was obtained from all volunteers. Blood samples were collected into polypropylene tubes containing sodium citrate from an antecubital vein with a 21-gauge needle, discarding the first 2 ml to avoid platelet activation. Blood was collected between 8:00 and 10:00 after overnight fasting. Samples were processed immediately. ACS patients were recruited at our cath-lab (both genders, age <80) from among those referred for coronary angiogram and subsequent percutaneous coronary intervention. Exclusion criteria were as follows: active treatment with β-blockers; any situation which might make it imprudent to administer an i.v.-β-blocker; asthma or chronic obstructive lung disease; bradycardio (HR <55 b.p.m.); heart failure or valvular heart disease; atrial fibrillation requiring antiarrhythmic therapy; renal failure with creatinine ≥ 2 mg per ml; liver disease with bilirubin ≥ 2 mg per ml; acute illness of any malignancy; pregnancy or nursing; body mass index ≥ 27 kg per m², previous severe adverse reaction to β-blockers; concomitant use of other antithrombotic drugs such as anticoagulants, dypiridamol, ticlopidine or cilostazol; treatment before the intervention with GP IIb/IIIa inhibitors, or need for nonsteroid anti-inflammatory drugs. Written informed consent was obtained from all patients enrolled. Blood samples were collected from a femoral artery catheter into polypropylene tubes containing sodium citrate, discarding the first 2 ml. Samples were processed immediately.

**Human neutrophil–platelet interactions evaluation.** Human citrated blood was diluted 1:5 in HEPES-Tyrode's* (5 mM hydroxyethylpiperazineethane-sulfonic acid (HEPES), 137 mM NaCl, 2.7 mM NaHCO₃, 0.36 mM NaH₂PO₄, 2 mM NaH₂PO₄, 2 mM CaCl₂, 5 mM glucose, bovine albumin 0.2%. pH = 7.4) and incubated with 0, 2 or 5 μM metoprolol for 10 min. Then, 5 ml of diluted blood was incubated with 5 μM epinephrine for 10 min. Unstimulated and epinephrine-stimulated samples were stained with PC5-conjugated anti-*CD14* anti *CD45*-FITC and anti-*CD61* PC7 for 20 min at room temperature in the dark. Erythrocytes were lysed for 10 min using Versalyse solution. Appropriate mouse isotype controls were used for each antibody. Flow cytometry analysis was performed with a Gallios cytometer (Beckman Coulter, Miami, FL, USA). Leukocytes were by *CD45*-FITC staining. Neutrophils identification was adapted from refs 34,35 and performed by morphological parameters (side scatter) and negative staining for *CD14*-PC5 but positive staining for *CD45*-FITC. Neutrophil–platelet conjugates were identified as bivariate histogram particles negative for *CD14*-PC5 and positive to *CD61*-PC7 (refs 36,37). The acquisition process was stopped after collection of 5,000 monocytes. Data are expressed as the percentage of neutrophil–platelet aggregates.

All experiments were conducted at the Hospital Universitario Clínico San Carlos, Madrid.

**Human platelet function evaluation.** Platelet aggregation was assessed using light transmittance aggregometry in PRP by the turbidimetric method in a four-channel aggregometer (Chrono-Log 490 Model, Chrono-Log Corp., Havertown, PA, USA) according to standard protocols. The PRP was obtained from citrated blood at centrifuge (800 r.p.m.) for 10 min and platelet-poor plasma was obtained after a second centrifugation (2,500 r.p.m.) for 10 min. PRP will be adjusted to 250,000 per μl with autologous plasma. PRP was incubated with metoprolol 2 and 5 μM or saline buffer for 15 min and then stimulated using epinephrine (5 μM). Light transmission was adjusted to 0% with PRP and to 100% with platelet-poor plasma for each measurement. Curves were recorded during 5 min and platelet aggregation was determined as the maximal percent change in light transmittance.

Platelet function was determined by assessing platelet activation as surface expression of activated GP IIb/IIIa (Becton Dickinson) and P-Selectin using flow cytometry. Whole blood from healthy donors were drawn into trisodium citrate tubes diluted with Hepes-tyrodes-buffer (0.2% BSA) to a final volume of 1:8:1 (blood: Hepes-tyrodes:citrate). Diluted blood was incubated with metoprolol 2 and 5 μM or saline for 15 min. Following activation with epinephrine (5 μM) samples were incubated for 20 min with polyclonal *PAC1*-FITC conjugated or PE-conjugated anti-*CD62P*. Appropriate isotype controls were used in each case. Platelet activation was expressed as the percentage of platelets positive for antibody binding. Platelets were gated on the basis of light scatter and *CD61* antibody expression. Activated platelets were defined as the percentage of expressing the activated confirmation of *PAC1* binding and P-selectin (*CD62P*). Data were expressed as the percentage of platelets positive for antibody binding. All experiments were conducted at the Hospital Universitario Clínico San Carlos, Madrid.

**Statistics.** Data are represented as mean ± s.e. of the mean (s.e.m.), and analysed using Prism software (Graph pad, Inc.) and Stata (Stata 12.0; StataCorp LP, College Station, TX, USA). Comparisons between two groups were performed by using the unpaired two-tailed Student's *t*-test or the nonparametric Wilcoxon–Mann–Whitney test as appropriate. Comparisons between more than two groups were performed by using the one-way ANOVA. The *P*-value was adjusted with the Holm Sidak's multiple comparison test. Multiple linear regression analysis was used to study the influence of metoprolol on MVO, adjusted for factors potentially affecting MVO such as sex, age, ischaemia duration, diabetes, use of thrombectomy or glycoprotein IIb/IIIa inhibitors. Test for linear trend after one-way ANOVA was used to study the relationship between LVEF at 6 months and MVO quartiles at 1 week. Power calculations were applied to obtain statistically significant at *P* values below 0.05 significant. *$P < 0.05$, **$P < 0.01$, ***$P < 0.001$.

**Study approval.** All studies in patients and volunteers were approved by the ethics committee of Hospital Clínico San Carlos, Madrid. Written informed consent was received from all participants before inclusion in the study.

**Data availability.** The data that support the conclusions of this study are available from the corresponding author on reasonable request.

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

## Acknowledgements

We are grateful to all the METOCARD-CNIC trial investigators and to Noemi Escalera for coordinating the clinical work and Maite D. Rodriguez for handling human samples (CNIC). Angel Macías and Barulio-Pérez-Asenjo (CNIC) performed the MRI studies. Noelia A-González (CNIC) supplied the LysM-GFP + mice. R.P. Khaton, J. Mateos and V. Zorita provided technical support and animal care. M. Casanova-Acebes gave advice on neutrophil isolation. J.M. Adrover for IVM technical advice. R. Doohan, A. Guijarro and A. Molina-Iracheta provided technical support in histology. A.M. Santos-Beneit helped with microscopy image analysis and movie editing. We also thank the CNIC Animal, Cellomics and Microscopy units for support. Simon Bartlett provided English editing. We are grateful to Eeva I. Soininen and Ana I. Castillo for their inconditional support and management of the grants supporting this work. This work was supported by a competitive grant from the Institute of Health Carlos III and the European Regional Development Fund (ERDF/FEDER) (PI10/02268, PI13/01979 & RD12/0042/0054). R. F.-J. was the recipient of non-overlapping grants from the Institute of Health Carlos III and ERDF/FEDER (Rio Hortega fellowship) and the Fundación Jesús Serra, the Fundación Interhospitalaria de Investigación Cardiovascular (FIC) and the CNIC (FICNIC fellowship). A.H. is funded by MINECO and ERDF/FEDER (SAF2015-65607-R) and Fundació La Marató-TV3 (120/C/2015-20153032). The CNIC is supported by the Spanish Ministry of Economy and Competitiveness (MINECO) and the Pro-CNIC Foundation, and is a Severo Ochoa Center of Excellence (MINECO award SEV-2015-0505). Borja Ibanez is the 2010 Princess of Girona awardee in Science.

## Author contributions

B.I. is responsible for the design of the entire study, assisted by J.G.-P. Experimental myocardial IR, peritonitis experiments, neutrophil migration assays and bone marrow transplants were done by J.G.-P., M.G., R.V.-G. and A.P.-G. Interpreted by B.I. Genotyping and transcripts expression evaluation were done by M.G., R.V.-G. and D.S.-R. Histological processing and evaluations were done by M.G., R.V.-G., R.B.-M. and J.G.-P. MVO evaluation in MRI studies from the METOCARD-CNIC trial were analysed by I.G.-L., R.F.-J., J.M.G.-R. and G.P. METOCARD-CNIC trial PIs: B.I and V.F. Correlation studies of WBC and MVO were done by J.M.G.-R. and J.G.-P. Interpreted by B.I. Intravital microscopy experiments and analysis were done by G.C., V.S. and A.S.dV. Interpreted by A.H. Human neutrophil–platelet interaction and platelet function experiments were done by E.B. and analysed by J.G.-P. Interpreted by A.F.-O and B.I. Statistical analyses were done by J.G.-P., J.M.G.-R. and R.F.-J. Manuscript was drafted by J.G.-P, and critically revised by A.H., V.F. and B.I. B.I. and J.G.-P are responsible for the final version of the manuscript, which was approved by all authors.

## Additional information

**Competing financial interests:** The authors declare no competing interests.

