## [Peer Review File · Nature Communications]

Reviewers' comments:

Reviewer #1 (expert in β 1AR and heart physiology)

Remarks to the Author:

In the present manuscript, the authors investigate the mechanism by which metoprolol reduces ischemia/reperfusion injury during acute myocardial infarction (AMI). In particular, the authors focused on the role of β 1AR activation of neutrophils and neutrophil/platelet aggregation as a crucial mechanism underlying reperfusion injury during AMI that is selectively inhibited by metoprolol therapy only when administered prior to reperfusion. This manuscript is generally well written and scientifically thorough, progressing from more basic studies to more mechanistic detail as appropriate. Throughout this study the authors used proper, validated controls and their conclusions are well supported by their data. Overall, the study would be of interest but there are some concerns that should be addressed.

Major Concerns:

1. The authors highlight the investigation of microvascular obstruction (MVO) as the mechanism by which neutrophil and platelet aggregates exacerbate reperfusion injury. In contrast to the CMR imaging, however, there appears to be no overt difference between the three H&E stained sections in Figure 3f. The authors should provide either a more detailed explanation of the visual manifestation of MVO in these images or, perhaps a different histological stain or the use of antibody labeling would provide a better visual representation of the damage.
2. Regarding all of the Evans blue/TTC stained representative images, the figure legend states that both sets of images are from the same hearts. If this is true, why does it appear to be opposite sides of the same section rather than the same side? Further, why does the area at risk tissue appear white following Evans blue, and why does the Evans blue staining disappear post-TTC staining leaving only red and white tissue. These inconsistencies should be thoroughly addressed in the text or methods section.

Minor Concerns:

1. In the first paragraph of the result section, the authors state that, "This significant effect was maintained after adjusting for factors potentially affecting MVO." Please provide a description of these factors and how the data was adjusted to account for them, since this information is currently excluded.
2. The last sentence of the I/R procedure methods has several typographical errors that should be corrected.
3. The Greek symbols were not maintained in the intravital microscopy methods section.
4. In Figure 3m the representative images suggest a significant decrease in infarct area after metoprolol therapy in contrast to the quantified data which shows no change. Further, the infarct area appears similar to the previous images (closer to 40% than 20%) in the control mice. Please confirm the quantified data and provide the most appropriate representative images.
5. In Figure 5 the legend, while stating i-k, seems to only complete the description of 5i, with 5j and 5k lacking appropriate text to address the graphs.
6. Figure 6a should provide a larger representative image area encompassing multiple vessels and then contain the insert or separate images zoomed in to one representative vessel.

7. In Figure 7a epinephrine is misspelled.

Reviewer #2 (expert neutrophils and platelets)

Remarks to the Author:

In this manuscript by Garcia-Prieto et al. the authors show that metoprolol exerts its cardioprotective mechanism in MI via direct effects on neutrophil beta1AR.

The protective mechanism of metoprolol in MI is well known but the report of Garcia-Prieto suggests an alternative mode of action: While previous studies mainly focused on cardiomyocytes, Garcia-Prieto et al. investigate the influence of neutrophils in this context. To elucidate the mechanism of metoprolol on neutrophils the authors investigate a patient cohort as well as mouse models of MI in combination with general genetic ablation of beta1AR and bone marrow transplantation. Neutrophil extravasation was further confirmed by sterile inflammation models using thioglycollate and complemented with in vitro experiments using human blood. While the study is of general interest to provide further information on the mode of action of metoprolol, the analysis of the underlying mechanism by Garcia-Prieto et al. left some unanswered questions:

Not only the number of neutrophils but also their state of activation and functional capacity is important to evaluate the effects observed. Did the authors determine the effect of metoprolol on other neutrophil effector functions? A previous report indicates effects of metoprolol on reactive oxygen species formation by neutrophils- is this effect responsible for the observed effects? The authors should provide more data on the underlying mechanism and discuss the current literature in more detail.

Does neutrophil influx influence subsequent extravasation of other leukocyte subpopulations e.g. monocytes, which essentially contribute to infarct size and healing processes?

The formation of heterotypic aggregates between neutrophils and platelets has been reported to be dependent on platelet activation. Previous studies investigated the effect of metoprolol on platelet function. However, these studies are not discussed in the manuscript nor is there any data provided that helps to understand the underlying mechanism of the observed metoprolol-mediated changes on platelet-neutrophil aggregates. While this study provides an interesting new hint, further evaluations are warranted to understand the consequences of metoprolol-mediated changes on neutrophil beta2 signaling in MI.

We thank editors and reviewers for their thorough revision to our work. We have addressed all issues raised by them and performed additional experiments when needed. Here we are answering point by point all issues raised and explain how the manuscript has been modified accordingly.

Reviewer #1

Reviewer: In the present manuscript, the authors investigate the mechanism by which metoprolol reduces ischemia/reperfusion injury during acute myocardial infarction (AMI). In particular, the authors focused on the role of β 1AR activation of neutrophils and neutrophil/platelet aggregation as a crucial mechanism underlying reperfusion injury during AMI that is selectively inhibited by metoprolol therapy only when administered prior to reperfusion. This manuscript is generally well written and scientifically thorough, progressing from more basic studies to more mechanistic detail as appropriate. Throughout this study the authors used proper, validated controls and their conclusions are well supported by their data. Overall, the study would be of interest but there are some concerns that should be addressed.

Authors: We appreciate the general comment from the referee.

Major Concerns:

Reviewer: The authors highlight the investigation of microvascular obstruction (MVO) as the mechanism by which neutrophil and platelet aggregates exacerbate reperfusion injury. In contrast to the CMR imaging, however, there appears to be no overt difference between the three H&E stained sections in Figure 3f. The authors should provide either a more detailed explanation of the visual manifestation of MVO in these images or, perhaps a different histological stain or the use of antibody labeling would provide a better visual representation of the damage.

Authors: We thank the reviewer for identifying that the H&E sections presented were not self-explainable. We agree that it is important to help readers understand the findings of the images. Following reviewer's suggestion, we have selected other H&E sections and inserted dashed lines and arrowheads to point areas of capillary obliteration (see below figure 3d of the revised version of the manuscript).

In addition we have inserted improved images of confocal microscopy of hearts with 3D reconstructed sections in Fig 3F (see below) and a supporting video of this confocal 3D images (supplemental video 1).

We believe these new images visually support the analyses done and will help readers to understand the clear effect of metoprolol on microvascular obliteration (mainly due to neutrophils-platelet aggregates)

Reviewer: Regarding all of the Evans blue/TTC stained representative images, the figure legend states that both sets of images are from the same hearts. If this is true, why does it appear to be opposite sides of the same section rather than the same side? Further, why does the area at risk tissue appear white following Evans blue, and why does the Evans blue staining disappear post-TTC staining leaving only red and white tissue. These inconsistencies should be thoroughly addressed in the text or methods section.

Authors: We have explained in more detail the Evans blue/TTC methodology in methods section of the revised manuscript. We are also referring to one of the previous publications from our group using the same methodology (Garcia-Prieto et al. *Basic Res Cardiol* 2014) in which it can be observed the same pattern for Evans blue/TTC staining.

In summary sections are exactly the same for Evans blue and for TTC staining. Evans blue is infused retrograde immediately after occluding the coronary artery. This results in blue staining of the irrigated remote area (area at risk persist uncolored). Then the heart is sliced and imaged. After imaging, each slice is incubated in TTC solution at 37°C for 15 min. TTC incubation results in strong red color of the alive myocardium and pale color of infarcted myocardium (negative for TTC). Sections are reimaged. This incubation results in Evans blue clearing out upon. This is why it is typical to present 2 Figs of the same slice (one after Evans blue and another after TTC). The reviewer is right that depending on the Evans blue solution concentration, blue color can be stronger and if incubated for less time in TTC, some blue color can be still be present. While this methodology seems interesting to be able to represent the 3 myocardial states in one single image (remote in blue, infarcted area at risk in white and alive area at risk in red), the accuracy for area at risk delimitation and mainly for infarct size quantification is not as good as with the lower Evans blue concentration. For this reason we decided many years ago to follow the concentrations explained in the methods section. We are not the only group doing this, rather many research groups use this technique and their images are similar to our (with Evans blue clearing out after TTC staining).

Regarding the question of whether Evans blue and TTC images were from the same section, yes they were. However, the rotation of the section was not exactly the same. We have inserted other sections in the revised version of the manuscript perfectly matching each other in rotation and with more contrast in Evans blue and TTC staining as suggested by the reviewer.

Minor Concerns:

Reviewer: In the first paragraph of the result section, the authors state that, "This significant effect was maintained after adjusting for factors potentially affecting MVO." Please provide a description of these factors and how the data was adjusted to account for them, since this information is currently excluded.

Authors: Following referee's comment, we are now describing the factors used for the model. The revised version of the manuscript now reads: "*This significant effect was maintained after adjusting for factors potentially affecting MVO by performing linear multiple regression analysis and including sex, age, ischemia duration, diabetes, and use of thrombectomy or glycoprotein IIb/IIIa inhibitors as covariates*".

Reviewer: The last sentence of the I/R procedure methods has several typographical errors that should be corrected.

Authors: Corrected, now it reads "*At the end of reperfusion, chest was closed and animals were kept with 100% O₂ and analgesia (buprenorphine S.C., 0.1 mg/kg) until the end of reperfusion*".

Reviewer: The Greek symbols were not maintained in the intravital microscopy methods section.

Authors: Corrected

Reviewer: In Figure 3m the representative images suggest a significant decrease in infarct area after metoprolol therapy in contrast to the quantified data which shows no change. Further, the infarct area appears similar to the previous images (closer to 40% than 20%) in the control mice. Please confirm the quantified data and provide the most appropriate representative images.

Authors: We thank the reviewer for noting this. The quantification has been checked and was done correctly, there are no differences in infarct size. We agree with the reviewer that the images were not the most representative. Following reviewer suggestion, we have now used more representative images (see figure below, Fig. 3o and 3p in the revised version of our manuscript).

5. In Figure 5 the legend, while stating i-k, seems to only complete the description of 5i, with 5j and 5k lacking appropriate text to address the graphs.

Authors: Corrected.

6. Figure 6a should provide a larger representative image area encompassing multiple vessels and then contain the insert or separate images zoomed in to one representative vessel.

Authors: Following the reviewer's suggestion, we have included an image with less zoom and then 2 additional zoomed images of the selected regions (see below, now being presented as Fig. 6b of the revised manuscript).

7. In Figure 7a epinephrine is misspelled.

Authors: Corrected.

Reviewer #2

Reviewer: In this manuscript by Garcia-Prieto et al. the authors show that metoprolol exerts its cardioprotective mechanism in MI via direct effects on neutrophil beta1AR.

The protective mechanism of metoprolol in MI is well known but the report of Garcia-Prieto suggests an alternative mode of action: While previous studies mainly focused on cardiomyocytes, Garcia-Prieto et al. investigate the influence of neutrophils in this context. To elucidate the mechanism of metoprolol on neutrophils the authors investigate a patient cohort as well as mouse models of MI in combination with general genetic ablation of beta1AR and bone marrow transplantation. Neutrophil extravasation was further confirmed by sterile inflammation models using thioglycollate and complemented with in vitro experiments using human blood.

While the study is of general interest to provide further information on the mode of action of metoprolol, the analysis of the underlying mechanism by Garcia-Prieto et al. left some unanswered questions:

Reviewer: Not only the number of neutrophils but also their state of activation and functional capacity is important to evaluate the effects observed. Did the authors determine the effect of metoprolol on **other neutrophil effector functions**? A previous report indicates **effects of metoprolol on reactive oxygen species formation by neutrophils**- is this effect responsible for the observed effects? The authors should provide more data on the underlying mechanism and discuss the current literature in more detail.

Authors: The reviewer raises a very important question. In our study we first observed an effect of metoprolol on the number of neutrophils in the post-ischemia/reperfusion myocardium in mice. Following the same rationale of the reviewer, we wanted to explore the function of the neutrophils in vivo. This was the reason for performing the intra-vital microscopy studies. Intra-vital microscopy is a highly accurate technique to document behavior (**function**) of neutrophils. We refer the reviewer to **the online video 3** in which it is clearly demonstrated that **metoprolol alters de function of neutrophils**, which cannot change their conformational structure and interaction with platelets to initiate infiltration of the tissue. We think this is a **robust demonstration that metoprolol alters neutrophils function** beyond the number of these reaching the myocardium.

The reviewer is right that it has been previously shown that metoprolol alters ROS production by neutrophils. However, it is unknown if the effect of metoprolol on this ROS production is mediated via β 1AR. Following reviewer suggestion, we have performed new experiments to address this issue. Oxidative burst assay was performed in neutrophils isolated from wild type and β 1KO mice. Cells were stimulated with W-peptide in the presence/absence of metoprolol. After incubation, ROS production was evaluated by flow cytometry. As can be seen the Figure below (Fig 4b in the revised version of our manuscript), metoprolol significantly attenuated ROS production of stimulated neutrophils. We show that this was β 1-mediated since metoprolol was not able to reduce ROS production in neutrophils from β 1KO mice.

In the present version of our manuscript we are including these new experiments and discuss the literature on this regard.

Reviewer: Does neutrophil influx influence subsequent extravasation of other leukocyte subpopulations e.g. monocytes, which essentially contribute to infarct size and healing processes?

Authors: Our study is focused on the early stages of reperfusion (early reperfusion injury). Infarct size (the main parameter evaluated to check protection afforded by metoprolol) was quantified by Evans blue/TTC staining at 24h. All hearts from animals sacrificed at 24h were used for this purpose and therefore were not available for microscopy (Evans blue retrograde perfusion at sacrifice precludes further fine histological evaluations). To evaluate early neutrophil-related (our main objective) events leading to infarct size

attenuation, 6h was chosen as the timepoint for harvesting hearts from additional mice for histological evaluations. With this design we cannot address the reviewer's question.

To address this question, we have done additional experiments (mouse ischemia/reperfusion sacrificed at 24h for additional histological exams). As can be seen in Figure below (Fig 3 g to j in the revised version of the manuscript), the infiltration of myeloid cells (LysM+) was significantly less in metoprolol-treated mice (data already presented before). At 24h, hearts from metoprolol-treated mice had a very mild increase of myeloid cells infiltration from 6 to 24h, while there was a sharp further increase of myeloid cell infiltration in hearts from control mice (Fig 3h). In panel 3i can be observed that the infiltration of neutrophils (LyM+/Ly6G+) followed the same pattern but the increase in control treated mice was less accentuated than for total myeloid cells. This suggest that from 6 to 24h, there was a significant increase in recruitment of other myeloid cells.

We have checked the infiltration of monocytes into the post-ischemia/reperfusion tissue at these 2 time-points (see figure below, suppl Fig 2 in the revised version of our manuscript). Confirming previous data, the density of monocytes increased more in hearts from vehicle treated mice than in hearts from metoprolol-treated ones. Given that animals were injected with metoprolol once before reperfusion, and half-life of this drug is short, we interpret that this effect seen in other myeloid cells could be secondary to the reduced infiltration of neutrophils and an attenuated "attract call" exerted by these, although a primary effect of metoprolol on these cells cannot be excluded. A detailed evaluation of the effect of metoprolol in other cell types deserves a dedicated study and currently is far from the scope of the present study.

In the present version of our manuscript we are also further presenting events occurring at 24h, as can be observed in the figure below (Fig 3k in the revised version of our work), and in a supplementary video (suppl video 2)..

Reviewer: The formation of heterotypic aggregates between neutrophils and platelets has been reported to be **dependent on platelet activation**. Previous studies investigated the effect of metoprolol on platelet function. However, these studies are not discussed in the manuscript nor is there any **data provided that helps to understand the underlying mechanism of the observed metoprolol-mediated changes on platelet-neutrophil aggregates**.

Authors: We appreciate the very wise comment from the reviewer. She/he is correct that it has been previously shown that some β blockers can exert a mild antiplatelet effect. To address the issue raised by the reviewer, we are presenting additional experiments in blood obtained from healthy volunteers in which we have evaluated the effect of metoprolol on platelet aggregation (see figure below, suppl Fig 8a in the revised version of our manuscript). We use the same metoprolol concentration as used in the neutrophil-platelet interaction assays. As can be seen in Figure, metoprolol did not result in any platelet inhibitory effect.

We also evaluated the effect of metoprolol on platelet markers associated with activation (P-Selectin and GO IIb/IIIa), see figure below (Suppl Fig 8b and 8c in the revised version of our manuscript). Again, effect of metoprolol on platelet activation markers was observed

These experiments strongly suggest that the effect of neutrophil-platelet interaction exerted by metoprolol is mainly due to a primary effect on neutrophils with no (little) involvement of platelet activation attenuation.

Reviewer: While this study provides an interesting new hint, further evaluations are warranted to understand the consequences of metoprolol-mediated **changes on neutrophil betaA2 signaling in MI.**

Authors: In this study we focused our attention into β 1AR. The evaluation of the effect of metoprolol on other β ARs, while extremely interesting and relevant, is far from the scope of our study and deserve a dedicated study. We had some indirect evidences showing that the effect exerted by metoprolol was not mediated by β 2AR: β 1KO mice (which have β 2AR expression in all cells) were not protected upon metoprolol administration (infarct size was unchanged). Similarly, the extreme effect of metoprolol on thyoglocolate-induced peritoneal migration of neutrophils was abrogated in β 1KO mice (which have β 2AR expression). However, given the comment from the reviewer, we have done additional experiments in this regards:

1) We have isolated neutrophils from β 2KO mice and evaluated the effect of metoprolol on neutrophil migration in the transwell assay. As can be seen in figure below (suppl Fig 5 in the revised version of our manuscript), neutrophils from β 2KO mice show an increased migration activity compared to WT neutrophils. Importantly, metoprolol significantly inhibited neutrophil migration in neutrophils isolated from β 2KO mice. These data reinforces the previously presented data and demonstrates that the effect of metoprolol on neutrophils is not β 2AR mediated.

2) In the former version of our manuscript we presented bone marrow transplants from wild type mice into β 1AR KO mice showing that the protective phenotype of metoprolol (lost in β 1KO mice) was restored when bone marrow from WT mice was transplanted.

In the present version of our manuscript we have included new experiments of opposite chimeras (β 1KO bone marrow into wild type). In this case the protective effect seen by metoprolol in wild type mice was lost when bone marrow from β 1KO was transplanted (Figure 4m of the revised version of our manuscript). These in vivo data strongly suggest the previously presented and shown that β 2AR expression was not involved in the protection afforded by metoprolol.

REVIEWERS' COMMENTS:

Reviewer #1 (Remarks to the Author):

The authors were very thorough in addressing the reviewer's concerns regarding their demonstration of microvascular obstruction and the methodology of their TTC staining, among other minor comments. The data are strengthened and clear. The authors have addressed all concerns, and the reviewer has no additional comments at this time.

Reviewer #3 (Remarks to the Author):

The authors convincingly show that neutrophil activation through beta1 adrenergic agonists promotes polarization, platelet attachment and microvascular obstruction. Metoprolol treatment largely reverses this. This is clinically relevant as shown by imaging data and a recent clinical trial. My comments are minor:

1. The authors say "Metoprolol acts during early phases of neutrophil recruitment by impairing the structural and functional rearrangements needed for productive engagement of circulating platelets with the uropod". Metoprolol seems to inhibit neutrophil polarization.
2. W-peptide-induced reactive oxygen species (ROS) production assay: W peptide should be introduced and referenced. Why is this thought to be better than fMLP?
3. Metoprolol was given as a bolus iv. What were the hemodynamic effects?
4. Why was epinephrin used in figures 4 and 7 and not a beta1-specific agonist?
5. Gating for CD14neg, CD45+, CD61+ will contain neutrophil-platelet aggregates, but this strategy is not specific. Excellent neutrophil markers exist.
6. Figure 1d: What the authors call "capillary obliteration" looks just like normal RBCs. The H&E is not a suitable method for this analysis.

We appreciate the thorough revision done to our resubmitted paper. Below we are addressing the remaining issues in a point by point rebuttal and detailing how the manuscript has been modified accordingly.

Q: Question raised by reviewer

A: Authors' response

Reviewer #1 (Remarks to the Author):

The authors were very thorough in addressing the reviewer's concerns regarding their demonstration of microvascular obstruction and the methodology of their TTC staining, among other minor comments. The data are strengthened and clear. The authors have addressed all concerns, and the reviewer has no additional comments at this time.

A: We appreciate the revisions done by this referee. These have helped improve the quality of the manuscript.

Reviewer #3 (Remarks to the Author):

The authors convincingly show that neutrophil activation through beta1 adrenergic agonists promotes polarization, platelet attachment and microvascular obstruction. Metoprolol treatment largely reverses this. This is clinically relevant as shown by imaging data and a recent clinical trial.

A: We thank the reviewer for her/his overall opinion of our work. We are particularly glad to read that reviewer feels our mechanistic study is clinically relevant.

My comments are minor:

Q1. The authors say "Metoprolol acts during early phases of neutrophil recruitment by impairing the structural and functional rearrangements needed for productive engagement of circulating platelets with the uropod". Metoprolol seems to inhibit neutrophil polarization.

A: Although metoprolol has a striking effect on the morphology and behavior of intravascular neutrophils, we believe that this occurs without impairing polarization. An important criteria to define polarization is the formation of two domains within seconds/minutes of neutrophil arrest (Hidalgo et al., Nat. Med. 2009): the uropod which can be tracked by L-selectin clusters (rather than homogeneous distribution); and the leading edge at the opposed side with protrusive and lamelopodial behavior. In our experiments we found that metoprolol did not disrupt formation of these domains, as illustrated in the images at the bottom panels of Fig.5g (L-selectin/uropod in yellow). Therefore, we conclude that metoprolol affects platelet capture, overall extension on the endothelium and the cells' dynamics (as shown in Figs. 5) without fully impairing polarization. This is the reason why we cautiously stated in the text that "metoprolol consistently disabled normal neutrophil polarization" to make this point. To clarify this better, we have changed this for "metoprolol consistently disabled the intravascular behavior of neutrophils without disrupting polarization within the activated vessels, resulting in ..." (page 9).

Q2. W-peptide-induced reactive oxygen species (ROS) production assay: W peptide should be introduced and referenced. Why is this thought to be better than fMLP?

A2: We agree with the reviewer on the suggestion and both introduction and reference for w-peptide have been included in the results and methods section of manuscript:

- In Results: "...by evaluation of chemotactic FPR activator-peptide, W-peptide, -induced reactive oxygen species (ROS) production assay."
- In Methods: "...cells were stimulated with the chemotactic FPR activator-peptide w-peptide 1 μ M (WKYMVM, 1799 Tocris)."

References #50 and #51 are now #30 and #31 respectively.

Regarding the different effect on oxidative burst between w-peptide and fMLP, we have not tested the latter. We used w-peptide after checking the literature in which some authors argued it was more effective than fMLP (see reference #30, where it can be read "...more effective than FMLP in the production of superoxide in human neutrophils..." , and reference 31#, where authors suggest using w-peptide when evaluating oxidative burst from murine neutrophils in section: "Measurement of PMN Oxidative Burst in Mouse Whole Blood").

Q3. Metoprolol was given as a bolus iv. What were the hemodynamic effects?

A3: This is an important comment. In the human studies, the dose of metoprolol was the one that is clinically approved, which was used in the clinical trial. Regarding the metoprolol dose used in the mouse model of ischemia reperfusion, at the beginning of the project we tested the hemodynamic effects of different i.v. bolus doses in WT mice at basal conditions. Heart rate and blood pressure (measured in the carotid artery) were documented after different concentration boluses. We concluded that 12.5mg/kg (50µL of metoprolol 10mM) dose was the highest dose without inducing a hemodynamic effect larger than 10% variation of heart rate and mean arterial pressure compared to pre-bolus. The 12.5mg/kg dose induced a 5% variation in blood pressure and 8% in HR.

To clarify this issue, we have included a sentence in page 16 of the revised version of our work: "*Dose of metoprolol was chosen after a dose response study, which identified this dose as the highest dose inducing a moderate effect on heart rate and blood pressure (i.e. <10% variation in both parameters from pre-dose), data not shown*".

Q4. Why was epinephrine used in figures 4 and 7 and not a beta1-specific agonist?

A: Our study was designed to evaluate the mechanism leading to a protective effect of metoprolol seen in the clinical scenario. In the acute myocardial setting there is a significant increase in circulating catecholamines. These catecholamines are unspecific (i.e. not beta1 selective). We designed experiments to be as similar to the clinical setting as possible. In addition, even selective beta1 agonists at certain concentration lose their specificity and binds to other beta adrenergic receptors. It is common practice in the field to use nonspecific adrenergic receptors and test specifically block (pharmacologically or genetically the receptor of interest). We decided to use epinephrine and test beta1 role by repeating experiments in neutrophils lacking this receptor (i.e. neutrophils from beta1 KO mice).

Q5. Gating for CD14neg, CD45+, CD61+ will contain neutrophil-platelet aggregates, but this strategy is not specific. Excellent neutrophil markers exist.

A5: We agree with reviewer that new markers are existing. We however used protocols adapted from Botto *et al.* Int J Cardiol 2007 and Xiao *et al.* J Am Coll Cardiol 2004 in order to identify leukocyte-platelet interactions. Neutrophil were identified by morphological parameters (side scatter), CD14 negative staining and CD45 positive staining. Methods from Van Velzen *et al.* and Giacomazzi *et al.* to identify platelets. All references have been now added to the manuscript.

- Botto N. *et al.* An increased platelet-leukocytes: a pathogenic role for "no-reflow" phenomenon? Int J Cardiol. 2007 Apr 12;117(1):123-30.
- Xiao Z. *et al.* Clopidogrel inhibits platelet-leukocyte interactions and thrombin receptor agonist peptide-induced platelet activation in patients with an acute coronary syndrome. J Am Coll Cardiol. 2004 Jun 2;43(11):1982-8.
- Van Velzen JF. *et al.* Multicolor flow cytometry for evaluation of platelet surface antigens and activation markers. Thromb Res. 2012 Jul;130(1):92-8.
- Giacomazzi A. *et al.* Antiplatelet Agents Inhibit the Generation of Platelet-Derived Microparticles. Front Pharmacol. 2016 Sep 16;7:314.

Still, we agree with the reviewer that we could have chosen newer more specific neutrophils markers. Still, given that all the experiments (human and animals) point to the same direction, we believe the selection of these neutrophil markers did not have a major role in reaching the conclusions of the experiments.

Following the reviewer comment, we have added a sentence in the manuscript stating that on limitation of these experiments is that there are more specific neutrophil markers that were not used in this study.

Page 13 (discussion section): *"In our flow cytometry studies we defined a positive neutrophil-platelet interaction as CD14neg, CD45+, CD61+ particle. It should be noted that there are more specific neutrophil markers and thus our selection is a limitation of these experiments. However, given that all data (human and mouse) point to the same direction, we think this limitation did not have an impact on the conclusions obtained in these experiments".*

Q6. Figure 1d: What the authors call "capillary obliteration" looks just like normal RBCs. The H&E is not a suitable method for this analysis.

A6: Every animal used for histological evaluation was subjected was gently infused through the vena cava with 10mL of PBS with heparin (50 U/mL) to avoid blood clots. Every accumulation of blood cells is assumed to be originated due to capillary obliteration.